# A 15-yr Circum-Antarctic Iceberg Calving Dataset Derived from Continuous Satellite Observations

Mengzhen Qi[1,3,5*], Yan Liu[1,3,5*], Jiping Liu[4], Xiao Cheng[2,3,5†], Yijing Lin[1], Qiyang Feng[1], Qiang Shen[6,7], Zhitong Yu[8]

[1] State Key Laboratory of Remote Sensing Science, and College of Global Change and Earth System Science, Beijing Normal University, Beijing 100875, China
[2] School of Geospatial Engineering and Science, Sun Yat-Sen University, Zhuhai 519082, China
[3] Southern Marine Science and Engineering Guangdong Laboratory, Zhuhai 519082, China
[4] Department of Atmospheric and Environmental Sciences, University at Albany, State University of New York, Albany, NY 12222, USA
[5] University Corporation for Polar Research, Beijing 100875, China
[6] State Key Laboratory of Geodesy and Earth's Dynamics, Innovation Academy for Precision Measurement Science and Technology, Chinese Academy of Sciences, Wuhan 430077, China
[7] University of Chinese Academy of Sciences, Beijing 100049, China
[8] China Academy of Space Technology, Qian Xuesen Laboratory, Beijing 100094, China

*These authors contributed equally to this work.

† *Correspondence to*: Xiao Cheng (chengxiao9@mail.sysu.edu.cn)

**Abstract.** Iceberg calving is the main process that facilitates the dynamic mass loss of ice sheets into the ocean, which accounts for approximately half of the mass loss of the Antarctic ice sheet. Fine-scale calving variability observations can help reveal the calving mechanisms and identify the principal processes that influence how the changing climate affects global sea level through the ice-shelf buttressing effect on the Antarctic ice sheet. Iceberg calving from entire ice shelves for short time intervals, or from specific ice shelves for long time intervals, has been monitored before, but there is still a lack of consistent, long-term, and high-precision records on independent calving events for all of the Antarctic ice shelves. In this study, a 15-yr annual iceberg-calving product measuring every independent calving event larger than 1 km$^2$ over all of the Antarctic ice shelves that occurred from August 2005 to August 2020 was developed based on 16 years of continuous satellite observations. First, the expansion of the ice-shelf frontal coastline was simulated according to ice velocity, and then the calved areas, which are considered to be the differences between the simulated coastline, were manually delineated, and the actual coastline derived from the corresponding satellite imagery, based on multi-source optical and synthetic aperture radar (SAR) images. The product provides detailed information on each calving event, including the associated year of occurrence, area, size, average thickness, mass, recurrence interval, and measurement uncertainties. A total of 1,975 annual calving events larger than 1 km$^2$ were detected on the Antarctic ice shelves from August 2005 to August 2020. The average annual calved area was measured as 3549.1 km² with an uncertainty value of 14.3 km², and the average calving rate was measured as 770.3 Gt/yr with an uncertainty value of 29.5 Gt/yr. The number of calving events, calved area, and calved mass fluctuated moderately during the first decade, followed by a dramatic increase from 2015/16 to 2019/20. During the dataset period, large ice shelves, such as

the Ronne-Filchner and Ross ice shelves, advanced with low calving frequency, while small- and medium-sized ice shelves retreated and calved more frequently. Iceberg calving of ice shelves is most prevalent in West Antarctica, followed by the Antarctic Peninsula and Wilkes Land in East Antarctica. The annual iceberg calving event dataset of Antarctic ice shelves provides consistent and precise calving observations with the longest time coverage. The dataset provides multi-dimensional variables for each independent calving event that can be used to study detailed spatial-temporal variations in Antarctic iceberg calving. The dataset can also be used to study ice-sheet mass balance, calving mechanisms, and responses of iceberg calving to climate change. The dataset is shared via National Tibetan Plateau Data Center, and entitled "Annual iceberg calving dataset of the Antarctic ice shelves (2005-2020)" with DOI: 10.11888/Glacio.tpdc.271250. In addition, the average annual calving rate of 18.4±6.7 Gt/yr of the calving events smaller than 1 km$^2$ of the Antarctic ice shelves, as well as the calving rate of 166.7±15.2 Gt/yr of the marine-terminating glaciers, were estimated.

## 1 Introduction

The ice shelves surrounding Antarctica's coastline play an important role in the stability of the Antarctic ice sheet and its mass balance. Iceberg calving is a process whereby the ice from a glacier or ice-shelf frontal edge is stripped away and enters the ocean. Iceberg calving accounts for approximately half of the net mass loss of all Antarctic ice shelves (Rignot et al., 2013;Depoorter et al., 2013). Enhanced iceberg calving can indirectly lead to ice shelf instability, which accelerates the outflow of tributary glaciers into the ocean, causing sea level rise (Berthier et al., 2012;Furst et al., 2016;Rignot et al., 2004). In-depth studies of the calving process are essential to accurately predict the impact of future climate change on ice shelves/sheets and sea levels.

Model simulations and remote sensing observations are two major tools used to study iceberg calving. The former focus on simulating the dynamic process of a calving front in response to atmospheric and oceanic forcings and stress within ice sheets. Different models are used to understand the evolution and changes of ice shelves (Hill et al., 2018;Lovell et al., 2017;Luckman et al., 2015;Miles et al., 2017). The latter focus on the monitoring and quantitative assessment of calved areas using remotely sensed data, which can be assimilated into ice sheet models to further improve the accuracy of model simulations (Massom et al., 2018;Pattyn and Morlighem, 2020).

Research on remotely sensed iceberg calving monitoring can be classified as having three main focuses: (1) observations of specific ice shelves or glaciers with high spatial resolution data, e.g., long-term monitoring of the Pine Island Glacier, Mertz Glacier Tongue, and Amery Ice Shelf (Bindschadler, 2002;Massom et al., 2015;Zhao et al., 2014); (2) observations made of larger regions with lower spatial and temporal resolution data, e.g., calving monitoring along the Antarctic Peninsula and Ross Sea coast (Cook et al., 2005;Cook and Vaughan, 2010;Fountain et al., 2017); and (3) circum-Antarctic calving front observations of specific years based on satellite image mosaics of the Antarctic coastline (Liu and Jezek, 2004;Liu et al., 2015;Scambos et al., 2007;Yu et al., 2019). The first two types of studies achieve the precise monitoring of calving events in specific ice shelves or small areas while the third type quantitatively assesses iceberg calving at the continental scale. Liu et al. (Liu et al., 2015)extracted 579 independent calving events for six years from the Envisat ASAR circum-Antarctic mosaic.

The authors obtained comprehensive, detailed iceberg calving observations at different scales through image matching and feature tracking, which made it possible to investigate calving patterns and mechanisms. Their work laid the foundation for the subsequent exploration of the physical triggers of small and large calving events (Medrzycka et al., 2016) and revealed the "self-organized critical systems" of glaciers and ice sheets at different calving scales (Åström et al., 2014).

The long-term and high-precision remote sensing observation of circum-Antarctic independent calving events not only describes the spatial and temporal features of iceberg calving but also provides fundamental data for further investigating calving mechanisms and estimating ice-shelf mass balance in response to climate change. In this study, we identify annual calving events through a combination of a velocity-based ice shelf front edge simulation and semiautomatic annual iceberg calving extraction. We further acquire the calved-area outline, location, year of occurrence, area, thickness, volume, mass, and recurrence interval of each calving event. Building on this, we develop a circum-Antarctic iceberg calving dataset. The dataset spans August 2005 to August 2020. Using this product, we analyse the spatial and temporal characteristics of iceberg calving for the last 15 years.

## 2 Data

### 2.1 Satellite imagery

Considering the relatively low calving frequencies measured in August of each year (Liu et al., 2013) and the time limitations of available satellite images, we define the annual calving recurrence interval as running from August of the current year to August of the following year. We know that it is difficult to create such a circum-Antarctic iceberg calving dataset based on a single satellite platform. To continuously monitor Antarctic iceberg calving for 2005 to 2020, multisource remotely sensed data are used in this study. We prioritize using SAR (Synthetic Aperture Radar) images for early August each year given that their quality is minimally affected by polar nights and cloudy days. For periods and areas for which SAR data are not available, optical images for close dates are used instead. Satellite images used in the development of this product include Wide Swath Mode (WSM) images from ENVISAT (Environmental Satellite) ASAR (Advanced Synthetic Aperture Radar) for 2005 to 2011 (downloaded from http://eogrid.esrin.esa.int/browse), MODIS (Moderate-resolution Imaging Spectroradiometer) 250 m Calibrated Radiances Product images (MCST, 2017) for 2012 to 2014 (downloaded from https://worldview.earthdata.nasa.gov/), the synthetic images of Landsat-8 OLI (Operational Land Imager) for bands 4 (630-680 nm), 3 (525-600 nm), and 2 (450-515 nm) for 2013 to 2020 (downloaded from https://www.usgs.gov/), and the Extra Wide Swath (EW) mode images of Sentinel-1 SAR for 2015 to 2020 (downloaded from https://www.esa.int/ESA). Detailed descriptions of these data are given in Table 1.

### 2.2 Supplementary datasets

Additional remote sensing data were also used to facilitate product development and analyses. MEaSURE InSAR (interferometric synthetic aperture radar)-based Antarctica ice velocity map version 2 (Rignot et al., 2011;Mouginot et al., 2012) is used to simulate the expansion of the ice-shelf frontal edge and locate calved areas. MEaSUREs Phase-Based Antarctica Ice Velocity Map Version 1 (Mouginot et al., 2019) and MEaSUREs Annual Antarctic Ice Velocity Maps 2005-

2017 Version 1 (Mouginot et al., 2017) are used for calving mass calculation for marine-terminated glaciers. MEaSUREs Antarctic Boundaries Version 2 (Rignot et al., 2013) is used for the ice shelf delineation and spatial analysis of the calving distribution. Two ice thickness datasets (Bedmap 2 and Bedmachine) (Morlighem et al., 2020;Fretwell et al., 2012) are used for calving thickness extraction and calving mass calculation for both calvings from ice shelves and marine-terminated glaciers.

105 The Reference Elevation Model for Antarctica (REMA) (Howat et al., 2019) is used for the uncertainty evaluation of the extracted thickness. The Antarctic daily surface melt dataset (Picard and Fily, 2006) is used to analyse the response of iceberg calving to ice sheet surface melting.

Detailed descriptions of each remote sensing product used are presented in Table 2.

**Table 1: List of satellite images used in the development of a circum-Antarctic iceberg calving product for 2005-2020**

| Satellite | Sensor | Product level | Agency | Swath | Revisit period in polar regions | Spatial resolution | Number of Images | Time range | Data acquisition |
|---|---|---|---|---|---|---|---|---|---|
| ENVISAT | ASAR (WSM) | L1B | ESA | 405 km | Less than 10 days | 75 ×75 m | 5,046 | 2005/08 - 2012/04 | http://eogrid.esrin.esa.int/browse |
| Sentinel-1 | SAR (EW) | L1 GRD | ESA | 400 km | Less than 6 days | 20×40 m | 3,780 | 2015/01 - 2020/08 | https://www.esa.int/ESA |
| Terra/Aqua | MODIS | L1B | NASA | 2,330 km | 1-2 days | 250×250 m | 168 | 2012/01 - 2014/12 | https://worldview.earthdata.nasa.gov/ |
| Landsat-8 | OLI | L1GT | NASA | 190 km | Less than 16 days | 30 ×30 m | 15,674 | 2013/11 -2020/8 | https://www.usgs.gov/ |

110 **Table 2: List of other remote sensing products used in the development of a circum-Antarctic iceberg calving product for 2005-2020**

| Dataset | Measurement methods (in ice-shelf areas) | Temporal coverage | Accuracy | Data Format | Agency | Data acquisition | Reference |
|---|---|---|---|---|---|---|---|
| MEaSUREs InSAR-Based Antarctica Ice Velocity Map Version 2 | InSAR | 1996-2016 | 1-17 m/yr | 450×450 m raster | NSIDC | https://nsidc.org/data/NSIDC-0484/versions/2 | (Rignot et al., 2017) |
| MEaSUREs Phase-Based Antarctica Ice Velocity Map, Version 1 | InSAR, speckle tracking | 1996-2018 | 0.1-10 m/yr | 450×450 m raster | NSIDC | https://nsidc.org/data/nsidc-0754/ | (Mouginot et al., 2019) |
| MEaSUREs Annual Antarctic Ice Velocity Maps | Speckle tracking, feature tracking | 2005-2017 | 1-32 m/yr | 1×1 km raster | NSIDC | https://nsidc.org/data/nsidc-0720 | (Mouginot et al., 2017) |

| | | 2005-2017, Version 1 | | | | | | |
|---|---|---|---|---|---|---|---|---|

| | | | | | | | | |
|---|---|---|---|---|---|---|---|---|
| MEaSUREs Antarctic Boundaries Version 2 | DInSAR | 1992-2015 | 25-250 m | Vector | NSIDC | https://nsidc.org/data/nsidc-0709/versions/2 | (Rignot et al., 2013) |
| Bedmachine | Hydrostatic equilibrium | 1970-2019 | 10 m | 500×500 m raster | NSIDC | https://nsidc.org/data/nsidc-0756 | (Morlighem et al., 2020) |
| Bedmap 2 | Satellite radar and laser altimetry, hydrostatic equilibrium | 1970-2000 | ~100 m, bias -13-53 m | 1×1 km raster | BAS | https://secure.antarctica.ac.uk/data/bedmap2/ | (Fretwell et al., 2012) |
| The Reference Elevation Model for Antarctica (REMA) | Stereo photogrammetry | 5/9/2015 ± 432 days | Less than 1 m | Digital Elevation Model | PGC | https://www.pgc.umn.edu/data/rema/ | (Howat et al., 2019) |
| Dataset of daily surface melt in Antarctica | Passive microwave radiometer (SMMR and SSM/I) | 1979-2018 | — | 25×25 km raster | UGA | http://pp.ige-grenoble.fr/pageperso/picardgh/melting/ | (Picard and Fily, 2006) |

\* Abbreviations. NSIDC for National Snow and Ice Data Center, BAS for British Antarctic Survey, PGC for Polar Geospatial Center at the University of Minnesota, and UGA for Université Grenoble Alpes.

## 3 Method

### 3.1 Processes of direct observation of annual independent calving event

115  An annual calving event occurs when an independent calved area has an outline that does not overlap or is spatially adjacent to other calving events occurring in the same year (even if it occurs on the same ice shelf), namely, the topology requires nonoverlapping and nonadjacent annual calved-area polygons for the specific year. Data generation involves the following three steps: preprocessing the data, extracting iceberg calving, and acquiring attributes (Figure 1). Each of these steps is discussed in the following sections. Besides, the consistency of multisource satellite imagery used in monitoring annual

120 iceberg calving has been validated.

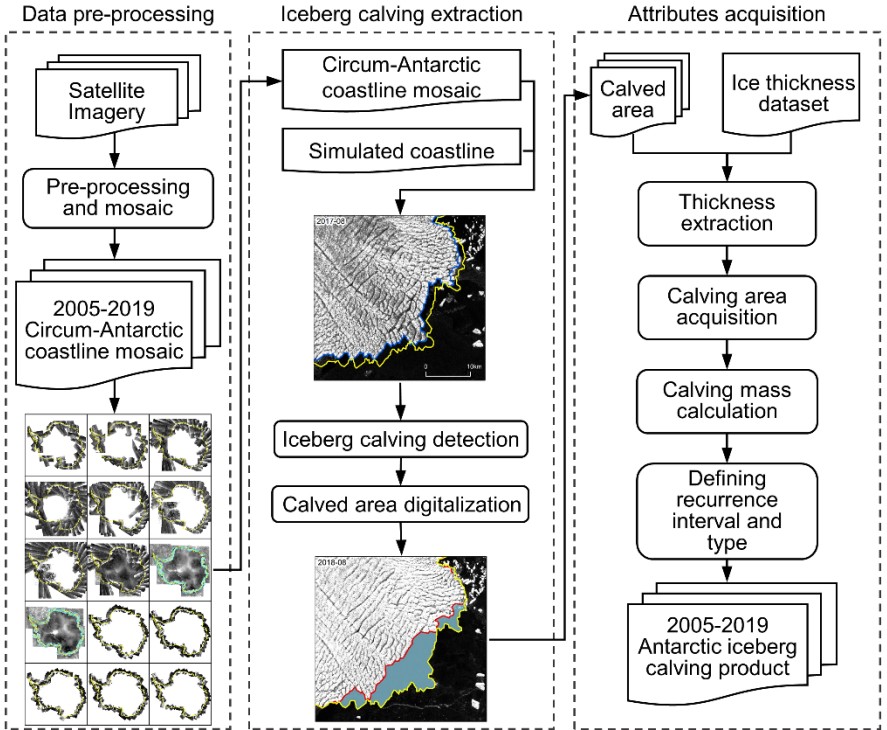

**Figure 1: Outline of our methodology. Satellite images are preprocessed to obtain the annual mosaic of the Antarctic coastline. Based on the circum-Antarctic coastline mosaic and corresponding simulated coastline, we extracted calved areas. Then, we acquired attributes such as thickness, area, volume, mass, and recurrence interval to produce an annual iceberg calving product for the Antarctic ice shelves.**

## 3.2 Data preprocessing of the remotely sensed image

Data preprocessing involves geocoding, geometric correction, and mosaic generation. SAR images for the first three days of each month of August are used preferentially to generate the circum-Antarctic coastline mosaic for the periods of 2005-2011 and 2015-2020. For 2012-2014, data vacancies were filled with images of the same sensors from close dates. For the mosaic for 2012, we used MODIS images for September combined with SAR images for April to facilitate detection. For 2013 and 2014 without SAR images, we chose both MODIS and Landsat-8 OLI circum-Antarctic coastline mosaics to extract iceberg calving. To reduce errors due to different imaging times, we overlaid the satellite image strictly by time order, namely, images taken on a date closer to 1st August should be on the upper layer. The preprocessing results of the remotely sensed data are shown in Figure 2, which provide good coverage of the Antarctic coastline and the frontal edges of ice shelves.

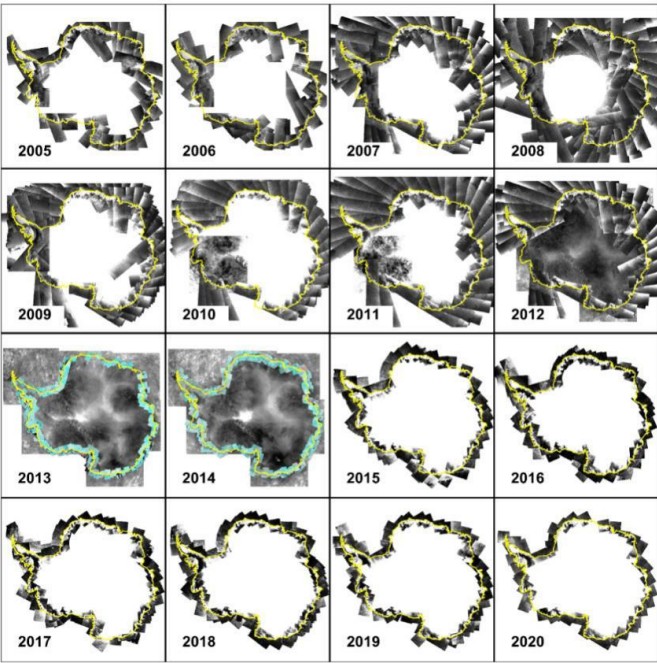

**Figure 2: Schematic showing the results of satellite imagery preprocessing. We mainly used Envisat ASAR images for 2005-2012, Landsat-8 OLI images for 2013-2014, MODIS images for 2012-2014, and Sentinel-1 SAR images for 2015-2020**

### 3.3 Iceberg calving extraction of independent calving events

To create the annual iceberg calving dataset for the Antarctic ice shelves, we simulated the expansion of the ice-shelf frontal edge and detected the calved areas based on satellite images. It is worth mentioning again that our iceberg calving extraction only included calving from ice shelves but did not include marine-terminating glaciers, and the boundaries of ice shelves are referenced from the MEaSUREs Antarctic Boundaries Version 2 released by NSIDC. We first manually digitalized the ice-shelf frontal line in August 2005, 2010, and 2015 as the input benchmark coastline. Then, the following steps were iterated for the extraction of each annual calving cycle with the methodology divided into two overarching tasks: velocity-based ice shelf front edge simulation and semiautomatic annual iceberg calving extraction (Qi et al., 2020).

**a) Velocity-based ice shelf front edge simulation.** We converted the vertices of the input coastline to obtain the set of coastline feature points for a specific year. Based on the velocity at the position of each coastline point, we calculated the movement of feature points over the duration of the given year. By lining up the moved feature points sequentially, a new coastline was derived, namely, the simulated coastline of the next year, as shown with yellow lines in Figure 3.

Additionally, we conducted a controlled experiment on the impact of different ice velocity products while simulating the next-year coastline. Fifty points on the high-flowing Pine Island Glacier were randomly selected as samples. We simulate their 11-yr movement using both the average ice velocity map (Rignot et al., 2017) and MEaSUREs Annual Antarctic Ice Velocity Maps for 2005-2017 (Mouginot et al., 2017). The results show that over the 11 years, the cumulative error between points moved under different ice velocity products by 0.15 km to 14.45 km with an average value of 3.96 km and a standard deviation

of 4.09 km. We assume that errors introduced by using the average ice velocity map to simulate the ice-shelf frontal edge of different years are acceptable.

For the non-calving area, theoretically, the simulated coastline should fit the real coastline shown in a remotely sensed image well, but due to the geographical bias of images and errors of the ice velocity product, some deviations between the directly obtained simulated coastline and actual coastline may occur. Therefore, before extraction, we first checked and

rectified the simulated coastline to ensure that it fits the actual coastline in non-calving areas. After manual correction, the extraction results were found to be of good accuracy.

**b) Iceberg calving extraction.** We manually rectified the simulated coastline to ensure that after rectifying, it fit the real coastline shown in the corresponding satellite images. Then, we obtained the actual coastline for the next year, which is shown as the red line in Figure 3. We extracted the enclosed area between the simulated coastline and the actual coastline to acquire

the calved area (the blue area in Figure 3). After extracting for one annual calving cycle, we checked topological relations at the continental scale for this year. We ensured that calved-area polygons did not intersect with each other and then obtained vectors for each calved area for the given year.

This iceberg calving extraction method employs a simple process and broad applications. The actual coastline modified from last year's extraction can be used as the input coastline of the next year's extraction; thus, we can provide time-continuous

iceberg calving monitoring and effectively avoid repetition and omission errors. Additionally, the semiautomatic operation offers incomparable precision and efficiency, greatly reducing the postprocessing workload.

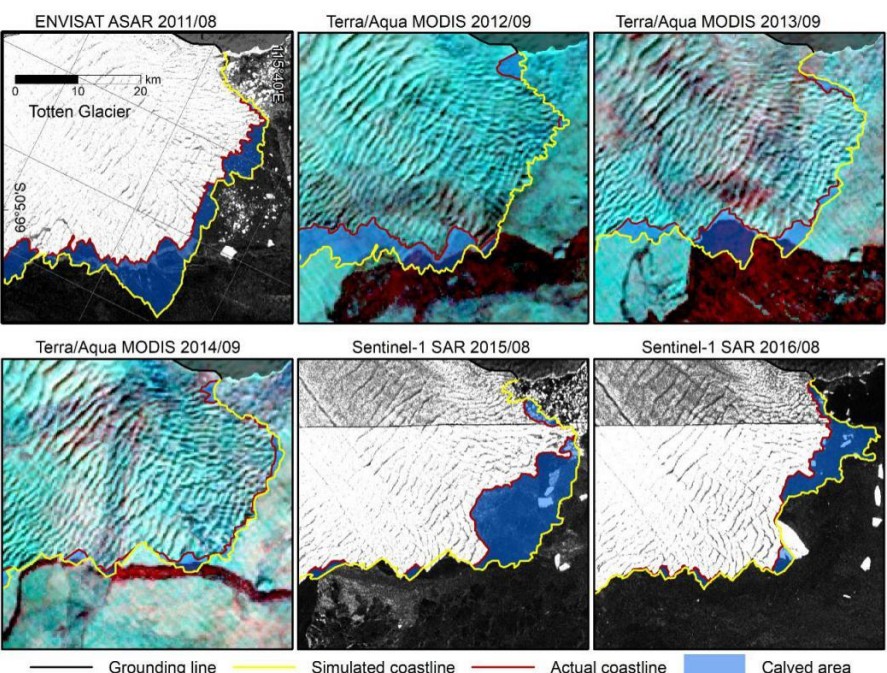

**Figure 3: Schematic of the calved area extraction method displaying different sources of satellite imagery used for annual iceberg calving extraction for 2011 to 2016. Red lines represent the actual coastlines. Yellow lines represent the simulated coastlines. Blue**
**areas represent the extracted calved areas.**

### 3.4 Attribute acquisition of independent calving events

For individual calving events, attributes include the area, calving scale, average thickness, mass, calving recurrence interval, and uncertainties of relevant parameters. Therefore, the acquisition of calved area and calved mass, uncertainties, and recurrence intervals are discussed in the following sections.

**3.4.1 Calved area and calved mass**

After acquiring vectors of the calved area polygons, we calculated their areas under polar projection. Then, these values were divided into four different scales: small-scale (1-10 km$^2$), medium-scale (10-100 km$^2$), large-scale (100-1,000 km$^2$), and extra-large-scale values (>1,000 km$^2$). We further obtained the average thickness of each calved area from the Antarctic ice thickness products (Bedmap 2 and Bedmachine). First, we masked out the ice-shelf zone thickness in Bedmap 2 and

Bedmachine. Second, we extracted the average thickness of each calving event from the masked ice thickness through step 1. Then, we checked the average thickness of all calving events. For missed and abnormal values (results show that they only account for a small proportion of the total), we moved the polygon backward along the ice flow to the calving front where there is thickness data coverage. After that, we re-extracted the average thickness of those calving events to make sure they are given appropriate thickness.

Based on area and thickness, the calving mass ($C$) was calculated from Eq. (1):

$$C = A_c \times \overline{H} \times \rho_{ice}, \tag{1}$$

where $A_c$ stands for the calving area and $\overline{H}$ represents the average thickness of the calved area. The standard value of ice density $\rho_{ice} = 917 \; kg/m^3$ was used for the calculation.

### 3.4.2 Uncertainty assessment

The uncertainties involved in the calculation of calving mass based on Eq. (1) include errors of calving area measurement, thickness extraction, and ice density. The uncertainty of the calving area is determined by the accuracy of the extraction method. Thickness uncertainty should be theoretically affected by top surface elevation measurements and firn depth correction; in reality, there are also uncertainties in thickness changes with time, according to hydrostatic equilibrium assumptions, and in the offsets in locations during extraction. In this section, we evaluate the main uncertainties encountered during the

development of the annual iceberg calving dataset.

**a) Calving area uncertainty.** Calving area uncertainty is mainly determined by the spatial location biases of calved-area outlines, which are related to both the original image resolution and the perimeter of the calved area. The equivalent perimeter width extracted by this method based on 75 m resolution images is 0.005 km (Qi et al., 2020); therefore, the uncertainty of the calving area ($U_A$) can be calculated from Eq. (2):

$$U_A = 0.005 \times l, \tag{2}$$

where $l$ represents the perimeter of each calving event (km).

**b) Thickness uncertainty.** The ice-shelf thickness dataset used in this product is derived from the hydrostatic equilibrium (Morlighem et al., 2020), which is written as Eq. (3):

$$H = (s - \delta)\frac{\rho_w}{\rho_w - \rho_{ice}} + \delta, \tag{3}$$

where $H$ denotes ice-shelf thickness. $s$ is the top surface elevation, namely, the height of the snow top. $\delta$ is firn depth correction, and $\rho_w = 1,027\ kg/m^3$ is the density of seawater.

Therefore, thickness uncertainty ($U_{\overline{H}}$) can be evaluated from Eq. (4):

$$U_{\overline{H}} = \overline{H} \times \sqrt{\frac{U_{s_c}^2}{s_c^2} + \frac{U_\delta^2}{\delta^2} + \frac{U_{\rho_{ice}}^2}{\rho_{ice}^2} + \frac{U_{\rho_w}^2}{\rho_w^2}}, \tag{4}$$

where $\overline{H}$ and $s_c$ represent the average thickness and average surface elevation of the calved area, respectively. $U_{s_c}$ is the

uncertainty of the calved-area surface elevation, $U_\delta$ is the uncertainty of firn depth correction, and $U_{\rho_{ice}}$ and $U_{\rho_w}$ represent the uncertainty of ice and seawater density, respectively.

For the calculations, 917 kg/m³ is used for $\rho_{ice}$ and 1,027 kg/m³ is used for $\rho_w$, and their uncertainties $U_{\rho_{ice}}$ and $U_{\rho_w}$ are valued at 5 kg/m³ (Griggs et al., 2011). $s_c$ was obtained from REMA with typical elevation errors of less than 1 m (Howat et al., 2019). Firn depth correction and its uncertainty were calculated from regional climate model RACMO2/ANT with a ratio

accounting for 8% (Pritchard et al., 2012).

**c) Calving mass uncertainty.** The calving mass of our dataset is derived from three components unrelated to and independent of each other. Thus, we used synthetic standard uncertainty to evaluate its accuracy. The mass deviation of a single calving event ($U_c$) is as follows Eq. (5), and the mass deviation for the year cycle ($\overline{U_C}$) can be calculated from Eq. (6):

$$U_c = C \times \sqrt{\frac{U_A^2}{A_c^2} + \frac{U_{\overline{H}}^2}{\overline{H}^2} + \frac{U_{\rho_{ice}}^2}{\rho_{ice}^2}}, \tag{5}$$

$$\overline{U_C} = \frac{\sqrt{\sum_{i=1}^n U_{C_i}^2}}{N}, \tag{6}$$

where $C$ and $A_c$ are the mass and area of individual calving events, respectively. $N$ is the number of years, and $n$ is the total frequency of calving events that occurred in $N$ years.

**3.4.3 Recurrence interval**

Calving recurrence means that a calving event with the same spatial scale reoccurs at the same calving front (Liu et al.,

2015), which are usually thought to be part of the natural cycle of advance and retreat of ice shelves. The recurrence interval

of a calving event, a measurement of the natural calving cycle, is defined as the year interval between the two recurrence calving events. To acquire this attribute, we performed the following work. First, we get the perimeter of each calving polygon through the function "Calculate Geometry" in ArcMap. Based on that, we calculated the average perimeter of all calving events at the same scale for 15 years. We defined the Buffer radii as half of the average perimeters at different scales rounded upwards to the nearest integer. The specific values used for this dataset are shown in Table 3.

**Table 3: Parameters used to define the calving recurrence interval.**

| Size | Perimeter (range)/km | Perimeter (average)/km | Buffer radius/km |
|---|---|---|---|
| Small-scale (< 10 km$^2$) | [4.0, 45.3] | 11.8 | 6 |
| Medium-scale (10-100 km$^2$) | [14.4, 136.2] | 37.4 | 19 |
| Large-scale (100-1,000 km$^2$) | [45.4, 184.0] | 93.6 | 47 |
| Extra-large-scale (>1,000 km$^2$) | [ 182.5, 479.5] | 310.2 | 155 |

After that, we used the function "Feature to Point" in ArcMap to get the center points of each individual calving polygon. For an input polygon, the location of the output point will be determined as its center of gravity. Then, we build buffers for each calving center point based on the radii calculated in the previous steps. For each calving event, we count the number of calving center points with the same scale that falls into its buffer. For buffers that fall into more than two points, the calving recurrence interval is defined as the total number of years (15) divided by the exact number of calving center points falling within. For buffers with only one point, the calving recurrence interval is defined as the greater value of time intervals between these calving events and boundary years (2005 or 2020).

**3.5 Consistency validation of multisource satellite imagery**

As mentioned above, a single satellite platform cannot accommodate long-time-series observations of circum-Antarctic calving events. Thus, multisource remotely sensed data are used in this study. To check whether the results derived from different sensors are similar, especially for the results derived from optical sensors and SAR, we performed the following verification.

For the year for which we have both SAR and optical images, we extracted circum-Antarctic annual iceberg calving using the same method based on different sources of remotely sensed imagery. We chose to repeat the calving extraction for 2016/17 through Terra/Aqua MODIS imagery and to compare it to the contemporaneous extraction results for our dataset derived from Sentinel-1 SAR imagery. We define area differences as the calving area obtained from MODIS subtracted from that obtained from SAR, and we define the calving perimeter as the calved-area perimeter obtained from SAR. Then, we analyse the area differences of the same calving events and calculate error-equivalent perimeter widths by dividing the area differences by the calving perimeter.

### 3.6 Estimation of the less than 1 km$^2$ calving from the Antarctic ice shelves

#### 3.6.1 Estimation method

Considering the huge workload and relatively small calving area contributing to the total calving area, we estimated the annual calving area and mass of the less than 1 km$^2$ calving of Antarctic ice shelves using the following equation:

$$A_{<1\ km^2} = (a + a^2 + a^3)\times A_{1-10\ km^2} \tag{7}$$

$$C_{<1\ km^2} = (a + a^2 + a^3)\times C_{1-10\ km^2} \tag{8}$$

where $a$ of 0.22 is the area ratio between the 0.1-1 km$^2$ calving and 1-10 km$^2$ calving estimated by Qi et al. (2020),$A_{<1\ km^2}$ and $A_{1-10\ km^2}$ are the calved area of the less than 1 km$^2$ caving and the 1-10 km$^2$, $C_{<1\ km^2}$ and $C_{1-10\ km^2}$ are the calved mass of the less than 1 km$^2$ caving and the 1-10 km$^2$, we have neglected higher-order terms in the expansion.

#### 3.6.2 Uncertainty assessment

The area uncertainty $U_{A_{<1\ km^2}}$ and the mass uncertainty $U_{C_{<1\ km^2}}$ of the less than 1 km$^2$ caving are calculated as follows:

$$U_{A_{<1\ km^2}} = (1 + 2a + 3a^2)\times \Delta a \times A_{1-10\ km^2} + (a + a^2 + a^3)\times U_{A_{1-10\ km^2}} \tag{9}$$

$$U_{C_{<1\ km^2}} = (1 + 2a + 3a^2)\times \Delta a \times C_{1-10\ km^2} (a + a^2 + a^3)\times U_{C_{1-10\ km^2}} \tag{10}$$

where $\Delta a$ of 0.05 is the standard deviation of $a$ estimated by Qi et al.(2020). $U_{A_{1-10\ km^2}}$ and $U_{C_{1-10\ km^2}}$ are the calculated uncertainties of 1-10 km$^2$ calving.

### 3.7 Estimation of the calving from the marine-terminating glaciers

#### 3.7.1 Estimation method

The calving rate of the marine-terminating glaciers is equal to the ice flux along their grounding lines. Ice flux comprises the flux gate width multiplied by ice velocity and ice thickness at the grounding line. The ice velocity and the ice thickness vary considerably from grounding line positions. Therefore, the grounding line is normally discretized to calculate the ice flux of each flux gate. Then, the calving rate $C_{marine-terminating}$ of the marine-terminating glaciers are calculated as follows:

$$C_{marine-terminating} = \sum_{i=1}^{n} H_i \times \vec{V}_i \times \vec{L}_i \times \rho_{ice} \tag{11}$$

where $H_i$ is the equivalent ice thickness of the flux gate $i$, $\vec{V}_i$ is the ice velocity along the ice flow direction, $\vec{L}_i$ is the fluxgate width along the ice flow direction, and $\rho_{ice}$ is the density of ice (917 kg/m$^3$).

#### 3.7.2 Uncertainty assessment

The calving mass uncertainty $U_{C_{marine-terminating}}$ of the marine-terminating glaciers are calculated as follows:

$$U_{C_{marine-terminating}} = C_{marine-terminating} \times \sqrt{\frac{U_{H^2}}{H^2} + \frac{U_{V^2}}{V^2} + \frac{U\rho_{ice}^2}{\rho_{ice}^2}} \tag{12}$$

where $U_H$ and $U_V$ stand for the uncertainties of ice thickness and ice velocity. For the calculations, 100 m is used for $U_H$ (Rignot, 2008;Rignot et al., 2011), and 17 m/yr is used for and $U_V$ (Mouginot et al., 2017). 917 kg/m$^3$ is used for $\rho_{ice}$ , and 5 kg/m$^3$ for its uncertainty $U_{\rho_{ice}}$ (Griggs et al., 2011).

## 4. Validation and Uncertainty

### 4.1 Consistency of multi-source satellite imagery

We extracted a total of 220 calving events from MODIS for 2016/17 covering a total area of 9,064.6 km$^2$. As shown in Table 4, both the total number of calving events and the total calved area are slightly lower than those derived from SAR imagery. The numbers of calving events at different scales extracted from the two sources of satellite images are similar. The frequency error mainly originates from small-scale calving, although it accounts for a small percentage of the total area. The calved area derived from MODIS at all four scales is underestimated compared with that from SAR, which might be a result of lower image quality for cloudy areas.

Table 4: Frequency and area distribution of different scale calving events derived from MODIS and SAR for 2016/17

|  | Scale | MODIS | SAR | Δ(MODIS-SAR) | Δ(MODIS-SAR)/SAR$_{Total}$ |
|---|---|---|---|---|---|
| Number of calving events | Small-scale (< 10 km$^2$) | 163 | 167 | -4 | -1.8% |
|  | Medium-scale (10-100 km$^2$) | 50 | 50 | 1 | 0.4% |
|  | Large-scale (100-1,000 km$^2$) | 6 | 6 | 0 | - |
|  | Extra-large-scale (>1,000 km$^2$) | 1 | 1 | 0 | - |
|  | Total | 220 | 224 | -4 | -1.8% |
| Total calved area (km$^2$) | Small-scale (< 10 km$^2$) | 511.0 | 563.0 | -52.0 | -0.6% |
|  | Medium-scale (10-100 km$^2$) | 1,441.0 | 1,478.2 | -37.2 | -0.4% |
|  | Large-scale (100-1,000 km$^2$) | 1,057.9 | 1,077.9 | -20.0 | -0.2% |
|  | Extra-large-scale (>1,000 km$^2$) | 6,054.7 | 6,141.0 | -86.3 | -0.9% |
|  | Total | 9,064.6 | 9,260.2 | -195.5 | -2.1% |
| Standard deviation of total calved area (km$^2$) | Small-scale (< 10 km$^2$) | 2.3 | 2.2 | 0.1 | 0.0 |
|  | Medium-scale (10-100 km$^2$) | 21.3 | 17.9 | 3.4 | 0.2 |
|  | Large-scale (100-1,000 km$^2$) | 93.4 | 91.9 | 1.5 | 0.0 |
|  | Extra-large-scale (>1,000 km$^2$) | 0.0 | 0.0 | 0.0 | - |
|  | Total | 397.2 | 402.8 | -5.6 | -1.4% |

The area of individual calving events extracted by MODIS is generally smaller. As the calving scale increases, errors caused by different data sources account for a lower percentage of the total calved area (Figure 4 (a)(b)(c)). The error-equivalent perimeter widths generally exhibit a normal distribution with a standard deviation of 0.15 km and a mean value of -0.06 km (Figure 4 (d)). Based on this, the errors introduced by multisource satellite data are acceptable.

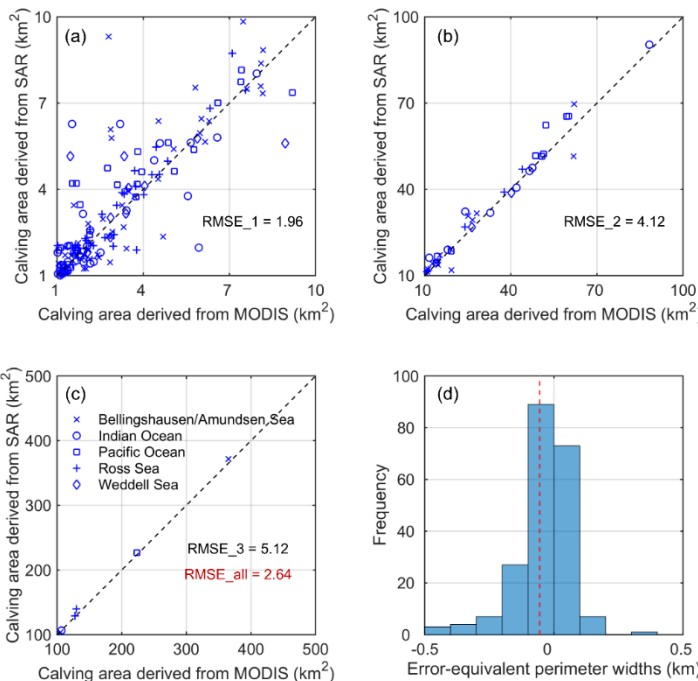

**Figure 4: Comparison of areas of individual calving extracted from MODIS and SAR for 2016/17. Panels (a), (b), and (c) show the small-scale, medium-scale, and large-scale calving events, respectively. Panel (d) shows the error distribution histogram of error-equivalent perimeter widths.**

## 4.2 Attribute uncertainties of independent calving events

We assessed the accuracy of the calved area, the calved-area thickness, and the calved mass attributes with Eq. (2), (4), (5), and (6).

The maximum area measurement uncertainty of a single calving event represented in this dataset was calculated as 30.7 $km^2$ with an annual average calved area uncertainty value of 14.3 $km^2$ and a standard deviation of 5.1 $km^2$. The calved area uncertainty is mainly determined by the perimeter of each single calving event. In the case of the same area, a long and narrow calving area has higher uncertainty than a square calving area. Thickness uncertainty is mainly attributed to firn depth correction. For individual calving events, thickness uncertainty ranges from 1.0 m to 67.7 m with a mean value of 18.5 m and a standard deviation of 9.1 m. The calved mass uncertainty is mainly determined by thickness uncertainty with a mean value of 29.5 Gt and a standard deviation of 23.6 Gt for 15 years, and its annual percentage fluctuates from 1.9% to 6.0% each year.

## 5. Temporal and spatial variations in Antarctic iceberg calving

### 5.1 Number, calved area, and calved mass of independent calving events

We identify 1,975 annual calving events covering areas larger than 1 $km^2$ occurring in the circum-Antarctic ice shelves from August 2005 to August 2020. The annual average number of calving events, the calved area, and the calved mass are 131.7 times, 3,549.1±14.3 km², and 770.3±29.5 Gt, respectively. The number of calving events, calved area, and calved mass

show high levels of year-to-year variability (Table 5), highlighting the need for longer records to determine long-term changes in ice shelves.

The number of calving events seems to be stable for the period of 2005/06-2015/16, fluctuating from 69 to 127, but it increases substantially in 2015/16 and fluctuates from 168 to 225 for the period of 2015/16-2019/20 (Figure 5(a)). The total calved area is anomalously low in 2006/07 compared to other years. Then, it increases in the following three years and especially in 2008/09 and 2009/10, during which two extra-large calving events occurred in the Wilkins Ice Shelf and Mertz Ice Shelf. Then, the total calved area decreased again in 2010/11 and fluctuated in 2010/11-2014/15. In 2016/17, the total calved area increased considerably to a maximum of 9,262.0 km$^2$ over the 15 years, during which an extra-large disintegration of the Larsen C Ice Shelf occurred. After that, we find the most dramatic reduction in 2017/18, with a total calved area of 1,386.3 km$^2$ reducing to a minimum during the observation cycle. In 2018/19, the total calved area rose slightly to a level close to that of 2005/06-2015/16, and in the following 2019/20, mainly contributed by the extra-large calving of Amery Ice Shelf in September 2019, the calved mass of that year reached the third-highest level of the 15-yr observation period. For annual calving mass, the maximum value appeared in 2016/17 at 1,832.6 Gt, and the minimum value in 2010/11 was recorded at 332.0 Gt. This fluctuating trend of calved mass is generally consistent with that of the calved area.

**Table 5: Annual distribution of the number of calving events, calved area, and calved mass for August 2005 to August 2020.**

| Year | Number of calving events | Calved area/ km$^2$ | Calved mass/ Gt |
|------|------|------|------|
| 2005/06 | 127 | 3,372.5±14.7 | 755.9±16.1 |
| 2006/07 | 98 | 1,702.5±12.2 | 402±6.2 |
| 2007/08 | 69 | 2,775.3±9.5 | 570.8±24.3 |
| 2008/09 | 113 | 4,341.3±15.2 | 704.4±18.7 |
| 2009/10 | 87 | 4,261.5±11.6 | 1,001.7±58.8 |
| 2010/11 | 83 | 1,707.6±9.6 | 332±6.4 |
| 2011/12 | 95 | 3,218.3±10.4 | 847.4±50.5 |
| 2012/13 | 119 | 2,932.2±12 | 762.7±37.9 |
| 2013/14 | 99 | 2,148±10.3 | 562.3±25.6 |
| 2014/15 | 73 | 2,262.4±8.7 | 552.5±13.8 |
| 2015/16 | 206 | 5,584.5±21.4 | 1,398.8±34.4 |
| 2016/17 | 224 | 9,260.2±26.5 | 1,832.6±94.9 |
| 2017/18 | 168 | 1,386.3±14.8 | 338.9±9.9 |
| 2018/19 | 225 | 2,806.4±17.9 | 732.9±23.2 |
| 2019/20 | 189 | 5,478.1±19.9 | 759.5±21.3 |
| Mean | 131.7 | 3,549.1±14.3 | 770.3±29.5 |
| Standard deviation | 55.5 | 2,042.1±5.1 | 399.4±23.6 |

## 5.2 Calved area and calved mass of the less than 1 km$^2$ calving from the Antarctic ice shelves and the calving mass from the marine-terminating glaciers

We assessed the annual calved area and calved mass of the less than 1 km$^2$ calving from the Antarctic ice shelves and the annual calving mass from the marine-terminating glaciers (Table 6). We indirectly estimated an average calved area of 92.7±27.8 km$^2$ and an average calved mass of 18.4±6.7 Gt/yr of the less than 1 km$^2$ calving from the Antarctic ice shelves. We

also take the calved mass of the marine-terminating glaciers into consideration by calculating the ice flux along grounding lines, which is about 166.7±15.2 Gt/yr. Therefore, the annual average calving rate of whole Antarctica is 955.4±51.4 Gt/yr.

**Table 6: Annual distribution of Calved area and calved mass of the less than 1 km² calving from the Antarctic ice shelves and the calved mass from the marine-terminating glaciers from August 2005 to August 2020.**

| Year | Calved area of the less than 1 km² calving / km² | Calved mass of the less than 1 km² calving / Gt | Calved mass of the marine-terminating glaciers / Gt |
|---|---|---|---|
| 2005/06 | 86.3±25.7 | 21.6±7.9 | 163.4±15.0 |
| 2006/07 | 64±19 | 15.1±5.5 | 168.6±15.2 |
| 2007/08 | 42.3±12.7 | 9.2±3.4 | 156±14.8 |
| 2008/09 | 68.6±20.4 | 15±5.5 | 174.4±15.4 |
| 2009/10 | 69.8±20.6 | 15.1±5.5 | 164.6±15.0 |
| 2010/11 | 48.5±14.4 | 8.4±3.1 | 172.7±15.4 |
| 2011/12 | 74±22.1 | 16.9±6.2 | 166.3±15.1 |
| 2012/13 | 95.9±28.5 | 15.3±5.6 | 163.4±15.0 |
| 2013/14 | 73.2±21.7 | 14.2±5.2 | 166.3±15.2 |
| 2014/15 | 43.2±12.9 | 8±2.9 | 163.8±15.1 |
| 2015/16 | 144.9±43.5 | 28±10.2 | 165.6±15.1 |
| 2016/17 | 158.2±47.7 | 31.2±11.4 | 175.7±15.5 |
| 2017/18 | 129.3±39 | 24.7±9 | - |
| 2018/19 | 171.5±51.6 | 34.6±12.6 | - |
| 2019/20 | 120.6±36.6 | 19.3±7 | - |
| Mean | 92.7±27.8 | 18.4±6.7 | 166.7±15.2 |
| Standard deviation | 42.3±12.8 | 8.1±3.0 | 5.5±0.2 |

345

## 5.3 Calving scale of independent calving events

The annual distributions of the number, total calved area, and total calved mass of calving events greater than 1 km² at different scales are shown in panels (a), (b), and (c) of Figure 5. Over the 15 years, the cumulative numbers of calving events of small-, medium-, large- and extra-large-scale events accounted for 72.6%, 23.5%, 3.5%, and 0.3%, respectively, and 350 frequencies increased exponentially as the scale decreased. The cumulative calved areas of the four different sizes accounted for 9.3%, 25.3%, 34.7%, and 30.6%, respectively. The distribution of calved mass is similar to that of the calved area.

The number of small-scale calving events accounts for a large percentage of total calving, especially in 2015/16-2019/20. The interannual variations in the number of small-scale calving events show obviously moderate variations. However, the area and mass of small-scale calving remain relatively stable and low. As the calving scale increases, interannual variations in 355 frequency become less significant; in contrast, interannual variations in area and mass become increasingly volatile. In some years, the number of calving events increased but calved area and mass remained stable or even decreased because more small-scale calving events made a limited contribution to the total calved mass and area. Thus, further studies must be conducted at different scales.

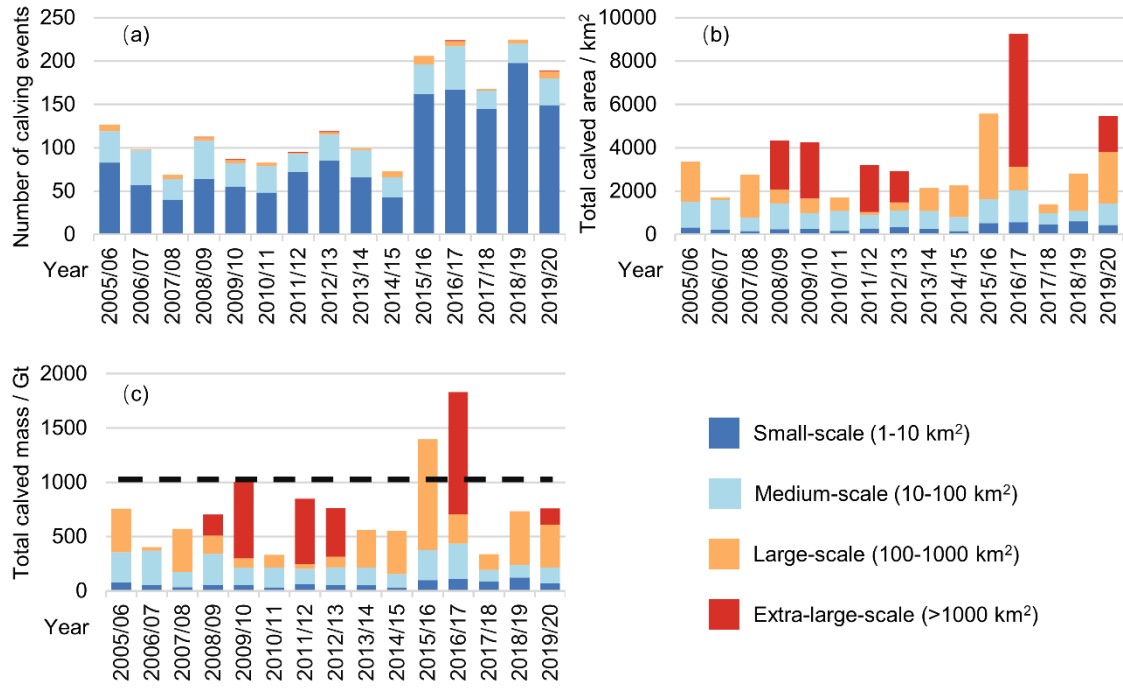

360 **Figure 5: Temporal distribution of annual calving events at different scales of Antarctic ice shelves from August 2005 to August 2020. Panels (a), (b), and (c) present the annual number of calving events, calved area, and calved mass at four scales, respectively. Horizontal dashed lines in Panel (c) denote the 1026 Gt/yr "steady-state" calving flux of ice shelves reported by Liu et al. (2015)**

## 5.4 Calving recurrence interval of independent calving events

The recurrence interval of calving provides additional qualitative information about the style of calving (Liu et al., 2015) and determines the suitable observation period for identifying ice shelf nonsteady-state behavior. For example, the rift-opening calving of the Amery Ice Shelf has reoccurred in 2019 since the last calving in 1963/64 (Li et al., 2020), detach along the boundary of isolated pre-existing rifts for decades. The observational records spanning many decades would be needed to determine its nonsteady-state behavior. In contrast, more frequent disintegration calving events are mainly caused by the hard to observe rapid basal crevasse propagation (Liu et al., 2015). The calving front retreat associated with these frequent calving events can be robustly identified over a short observation period due to the shorter recurrence intervals. In other words, the calving events with shorter recurrence intervals are more sensitive to current climate change.

Figure 6 (a) shows the calving recurrence interval is little related to calving scales of caving. The two extra-large-scale ($> 1,000\ km^2$) calving events reoccurred on the Thwaite Glacier during our observed period indicating its distinct retreat, while the other four extra-large-scale events from the Larsen C, Wilkins, Totten, and Amery Ice shelves did not reoccur. Figure 6 (b) shows that 76% of the total number of calving events reoccurred during the observed period (i.e., their recurrence intervals of calving are less than 8 yr), which suggests that the annual calving number is likely to be an indicator of the response of calving to climate change. Nearly half of the cumulative calved area from the events with the recurrence intervals greater than 8 yr

(i.e., the events only occurred once during the observed period) suggests that the annual calved area is not suitable for identifying the nonsteady-state behaviors of some ice shelves.

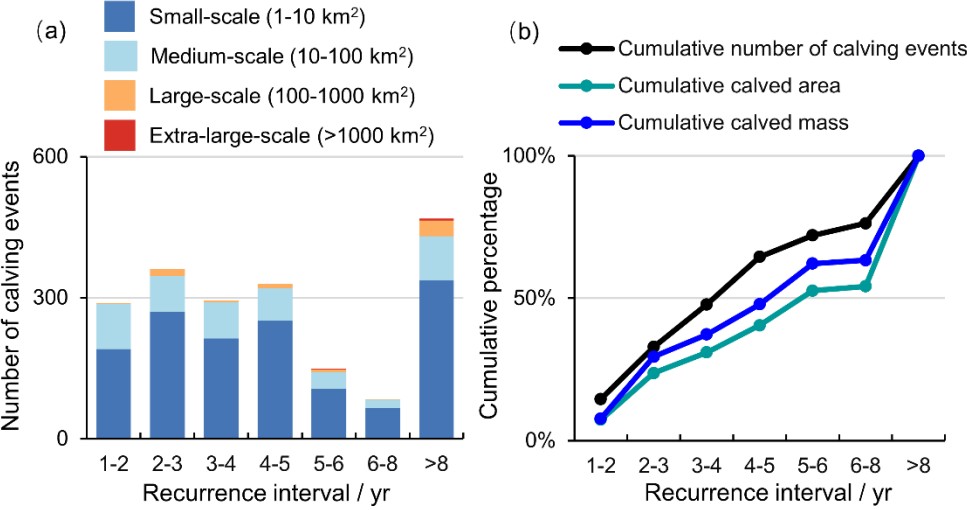

**Figure 6: Distribution of calving events with different recurrence intervals. Panel (a) shows the cumulative number of calving events at different scales. Panel (b) shows the cumulative percentages of the cumulative number of calving events, the cumulative calved area, and the cumulative calved mass.**

### 5.5 Spatial distribution of independent calving events

Figure 7 shows the spatial distribution of annual calving events at different scales from 2005/06 to 2019/20. Small- and medium-scale calving widely appeared in the Antarctic Peninsula, in West Antarctica, and on Wilkes Land in East Antarctica with interannual variations mainly found in Queen Maud Land in East Antarctica from 2011/12-2015/16. In 2011/12-2015/16, small-scale calving events were largely distributed in West Antarctica and sparsely occurred in East Antarctica. Large-scale calving events appeared quite randomly, usually in medium-sized ice shelf regions of the Antarctica Peninsula and West Antarctica. Extra-large-scale calving events only occurred twice in the Antarctica Peninsula, twice in West Antarctica, and twice in East Antarctica.

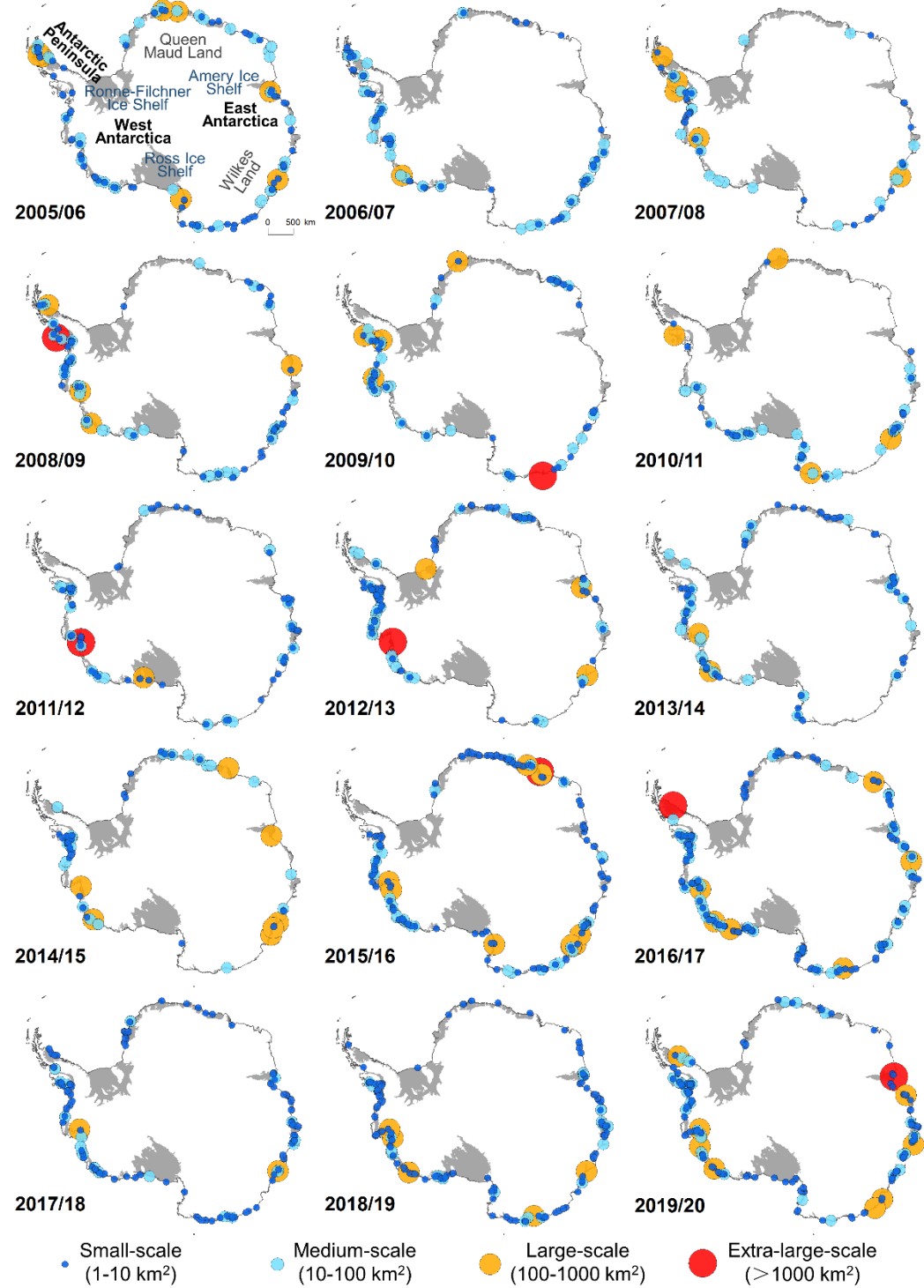

**Figure 7: Spatial distribution of annual calving events at different scales from August 2005 to August 2020.**

Figure 8 shows the average calving rates of the different Antarctic ice shelves from August 2005 to August 2020. We find that the calving mass of Antarctica is mainly affected by the iceberg calving of small ice shelves rather than that of larger ice shelves. Of these, the cumulative calving masses of two major ice shelves, the Ronne-Filchner Ice Shelf and the Ross Ice Shelf, are negligible over the 15-yr observation cycle. The Amery Ice Shelf and the Larsen C Ice Shelf in the Antarctic Peninsula, the third- and the fourth-largest ice shelf in Antarctica, had a very low calving rate except in the case of the extra-large disintegration event that occurred in September 2019 and July 2017, respectively. Additionally, some large ice shelves in Queen Maud Land show a low calving mass, while in some years, there were few calving events along the entire coastline (Figure 7).

In contrast, small- and medium-sized ice shelves, widely found along the circum-Antarctic coastline, exhibit a higher calving rate (Gt/yr). Among them, the Thwaites Ice Shelf, Pine Island Ice Shelf, and Getz Ice Shelf in West Antarctica show calving rates of 108 Gt/yr, 91 Gt/yr, and 52 Gt/yr, respectively. These are followed by the Mertz Ice Shelf and Totten Ice Shelf in East Antarctica with calving rates of 52 Gt/yr and 35 Gt/yr, respectively. Notably, we detected calving events at Totten Ice Shelf every year during the observation period, unlike the average calving mass of the Mertz Ice Shelf is mainly contributed by an extra-large disintegration event covering more than 2,500 km$^2$ that occurred in February 2010.

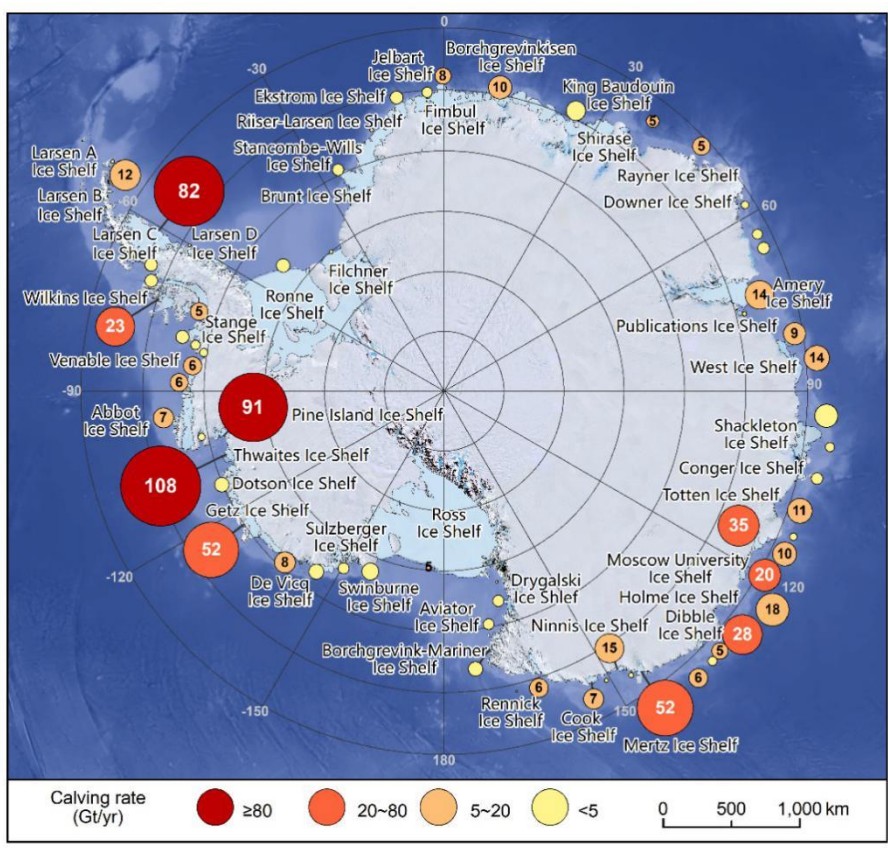

**Figure 8: Spatial distribution of average calving rate (Gt/yr) of Antarctic ice shelves from August 2005 to August 2020.**

## 6. Discussion and conclusion

The annual iceberg calving dataset of the Antarctic ice shelves (2005–2020) is the first that provides consistent and precise calving observations with the longest time span of 15 years. It not only directly reflects the quantitative characteristics and spatial distribution of Antarctic iceberg calving, but it also provides multi-dimensional variables of each independent calving event. This dataset can be used as fundamental data for subsequent studies on ice-sheet mass balance, calving mechanisms, and their responses to climate change.

The interpretation of calving records spanning 12 orders of magnitude from 1 to $10^{12}$ m$^3$ has demonstrated that the probability of calving events obeys a particular pattern whether they are small or large events—much like the Gutenberg-Richter law for earthquakes (Åström et al., 2014). Thus, the fine-scale and continuous observation of calving can be used to investigate how close particular glaciers are to their critical point, and thus how sensitive they may be to near-future changes in climatic and geometric conditions. However, finer-scale direct observation is greatly limited by the accessibility of high-resolution remotely sensed imagery and significant manual overhead. Our observations provide records of calving volumes ranging from $10^8$ to $10^{12}$ m$^3$ of Antarctic ice shelves.

The calved-area uncertainty of our direct observation (Qi et al., 2020) is dependent on the spatial resolution of the imagery, uncertainty of velocity data, and the perimeter-to-area ratio of the calved area. In the case of the same area, a long and narrow calving area has higher uncertainty than a square calving area. The relatively low-spatial-resolution satellite imagery used in this work and the characteristic of a long and narrow calving area are the main reasons that this method is not suitable for high-accuracy calving observation of marine-terminating glaciers. The trade-off between workload and uncertainty reduction is another consideration in choosing the minimum spatial scale of calving observation. With the calving scale decreasing from 100 km$^2$, the number of annual calving events increases exponentially, which means that the monitoring workload also increases exponentially (Qi et al., 2020). Although direct calving observation has the minimum valid extraction area of 0.05 km$^2$ based on 75-m SAR resolution images (Qi et al., 2020), it is uneconomical to observe calving events of less than 1 km$^2$ using exponentially increasing manual workload to reduce slightly the uncertainty of the total calving-rate estimation. This is why in the present work the calving area and mass of calving events of less than 1 km$^2$ of the Antarctic ice shelves were estimated based on observation-area ratio and direct observation of 1–10 km$^2$ calving events.

The total circum-Antarctic iceberg calving rate of 955.4±51.4 Gt/yr between 2005 and 2020 observed and estimated in the present study is less than the steady-state iceberg calving fluxes of 1,265 Gt/yr estimated by Rignot et al. (2013) and 1,321 Gt/yr estimated by Depoorter et al.(2013), respectively. The steady-state calving flux is the calving flux necessary to maintain an assumed steady-state calving front for a given set of ice thicknesses and velocities along the ice-front gate (Rignot et al., 2013; Depoorter et al., 2013). Such "flux-gate" calving calculations for the marine-terminating glaciers are suitable. Our estimated calving rate of the marine-terminating glaciers, 166.7±15.2 Gt/yr, is very close to that reported by Rignot et al.(2013), i.e., 176 Gt/yr. However, such "flux-gate" calving calculations for ice shelves are inevitably biased as they underestimate iceberg calving for retreating ice shelves or overestimate it for advancing ice shelves. Our observed average calving rate of 770.3±29.5 Gt/yr from calving events larger than 1 km$^2$ between 2005 and 2020 is slightly greater than the average rate of 755

Gt/yr between 2005 and 2011 (Liu et al., 2015), which is contributed by two distinct high calving rates of 1,398.8 Gt/yr in

2015/16 and 1,832.6 Gt/yr in 2016/17, respectively. The average calving rate of 788.7±36.2 Gt/yr of all of the Antarctic ice

shelves between 2005 and 2020 is the sum of 770.3±29.5 Gt/yr and the estimated average calving rate of 18.4±6.7 Gt/yr from

calving events less than 1 $km^2$, which is less than the steady state calving fluxes of 1,089 Gt/yr estimated by Rignot et al.(2013)

and 1,026 Gt/yr estimated by Liu et al.(2015), respectively. Thus, the Antarctic ice shelves are growing in extent.

Observations show that enhanced iceberg calvings have primarily been attributed to varying atmospheric and oceanic

conditions (Shepherd et al., 2003;van den Broeke, 2005;Scambos et al., 2009;Braun and Humbert, 2009;Liu et al.,

2015;Massom et al., 2018). Previous studies have revealed that ocean-driven thinning enhances iceberg calving and the retreat

of Antarctic ice shelves based on the first record of all icebergs larger than 1 $km^2$ calving from all of the Antarctic ice shelves

between 2005 and 2011 (Liu et al., 2015). Here, the time series of this dataset has been extended from 6 to 15 years. The

calving probability of Antarctic ice shelves indicated by the number of calving events has obvious inter-annual variation during

our observation period (Figure 6). Because 76% of the calving events have recurrence intervals of less than 8 yr, the annual

variation of the number of calving events thus probably reflects the calving response to current climate variability. This

provides an opportunity to examine the potential associations between iceberg calving and remote and local climate forcings.

Here, we show two examples from our preliminary analysis. First, Figure 9 (c) and (a) shows the relationship between

the number of calving events and the oceanic Niño index. Remotely, El Niño leads to anomalous increases in sea surface

temperature and Antarctic ice sheet temperature. We found that a strong El Niño might lead to an increase in the number of

calving events of Antarctic ice shelves, and there has been intensified iceberg calving since then. Second, Figure 9 (c) and (b)

shows the correlation between iceberg calving and ice sheet surface melting. Locally, atmospheric warming intensified ice

sheet surface melting, resulting in increased meltwater, which may trigger the expansion of rifts and crevasses and finally

enhance iceberg calving. Based on this dataset, we found significant positive correlations between the maximum daily surface

melting area and the number of calving events (r=0.76, p=0.003).

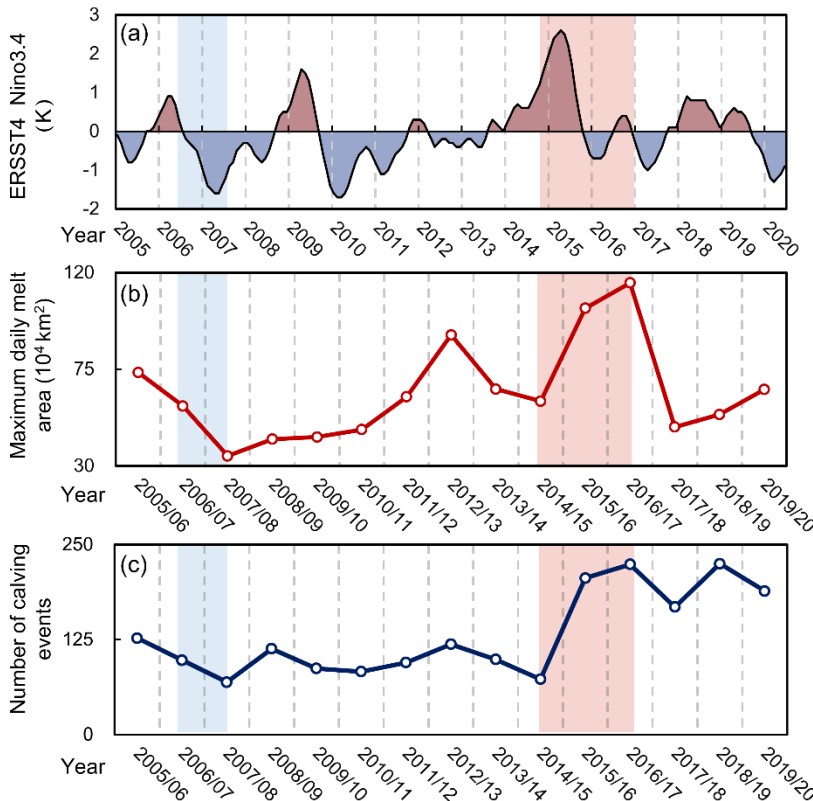

**Figure 9: Relationship between annual iceberg calving distribution for 2005 to 2020 and (a) oceanic Niño index data from https://ggweather.com/enso/oni.htm and (b) maximum daily ice sheet surface melting area data from http://pp.ige-grenoble.fr/pageperso/picardgh/melting/**

## 470    7. Conclusion and Data availability

The developed iceberg calving product applies a 15-yr calving distribution with year, length, area, scale, thickness, volume, mass, recurrence interval, and measurement uncertainty attributes for each calving event. The product applies an annual temporal resolution, and its spatial resolution is set to 1 km². The dataset is stored in Shapefile format, shared via the National Tibetan Plateau Data Center (http://data.tpdc.ac.cn/en/), and entitled "Annual iceberg calving dataset of the Antarctic

ice shelves (2005-2020)" with DOI: 10.11888/Glacio.tpdc.271250 (Qi et al., 2021).

**Author Contributions.** X.C, Y.L, and M.Q. conceived of, designed, and conducted the experiment. Y.L and M.Q. contributed to the research framework and helped develop the methodology. M.Q and Y.L. performed the data analysis. M.Q, Y.L, J.L, and Q.S. contributed to analysing the results. M.Q and Q.F. contributed to the data processing. All authors

contributed to the discussion and writing of the manuscript.

**Competing interests.** The authors declare that they have no conflict of interest.

**Acknowledgments.** We greatly thank European Space Agency (ESA) for providing the Sentinel-1 and Envisat imagery, we

thank the National Aeronautics and Space Administration (NASA) and the United States Geological Survey (USGS) for

providing the Landsat and MODIS data, and we truly appreciate the National Snow and Ice Data Center (NSIDC) for

providing the ice velocity and ice-sheet boundary products. We thank British Antarctic Survey (BAS) for providing the

Bedmap 2 and Polar Geospatial Center at the University of Minnesota and UGA for providing the Reference Elevation

Model for Antarctica(REMA).


**Financial support.** This research was funded by the National key research and development Program of China (Grant No.

2016YFA0600103 and Grant No. 2018YFA0605403), National Natural Science Foundation of China (Grant No. 41925027),

and Qian Xuesen Lab. - DFH Sat. Co. Joint Research and Development Fund.

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
