# Peer review of "A 15-yr Circum-Antarctic Iceberg Calving Dataset Derived from Continuous Satellite Observations"

_Earth System Science Data, 2020_

## Referee Comment (RC2)

Comments for ESSD-2020-340

**General comments:**

The manuscript presents a 14-year Antarctic Iceberg calving dataset using satellite remote sensing dataset. The calving events were detected based on the manual digitalized coastline and the simulated coastline using ice velocity. Then the number of calving events, calving areas, mass, and their uncertainty are derived. Afterwards, the spatial distribution and average calving rates for different ice shelves were obtained. I think the dataset is of great significance for relevant studies in Antarctica. However, my **main concern** for the dataset and discussions is regarding the uncertainty brought by small scale calving events and the accuracy of ice velocities.

In the manuscript, the calving areas and events were based on the difference between digitalized coastline and the simulated coastline using ice velocity. The authors also indicated that the velocity product has an error of 3.96+-4.09 km, which should have a significant influence on small scale calving events (1-10km$^2$). However, the uncertainty introduced by ice velocity was not analyzed in the uncertainty assessment, and those calving events with high-uncertainty were also not excluded from the following discussions and analysis. These may raise wrong conclusions. For example, would this affect the conclusion that the authors indicated that after 2015, there are more obvious calving events than the year before, and majority of them were small scale calving events. You may estimate the confidence level according to the events.

**Minor comments:**

1. P2, L55, the authors grouped the state of arts by different spatial resolution, not by the detection method?

2. P6, Section 3.2, it's not clear here why August 2005, 2010 and 2015 were used as the input benchmark. According to the descriptions in this section, every year's actual coastline modified from last year's extraction was used as the input of next year.

3. P8, L 178, to my knowledge, Bedmap2 may not provide thickness data in some of the coast areas, how do you deal with such situation.

4.P 10, Section 3.3.3, this section is not clear, how to define the center point, the perimeter

of a calving area? Line 224-L228, this paragraph is confusing.

5. P10, L230, the iceberg calving events were divided into two types, high frequency and low frequency; it seems contradictory with calving frequency (Table 4). Is calving frequency means the number of calving events in every year?

6. P11, Table 4, please also include the standard deviation for the calving areas.

7. P 12, add standard deviation in section 4.2.

8. The calving events with high uncertainty should be excluded or discussed separately in Section 5.

---

## Community Comment (CC1)

[supplement omitted: unrelated document]

---

## Author Comment (AC2)

Response to Professor Chad A. Greene:

We sincerely thank the reviewer for the insightful and constructive comments and important suggestions. These comments were very helpful for revising and improving our paper. We have endeavored to address all of the comments and to improve the manuscript to the best of our ability. Our point-by-point responses are detailed below.

**Comment 1:**

(Page 1) **Under reported value of calving flux**: The 771.1 Gt/yr value of calving flux presented here is significantly lower than the 1265 Gt/yr value reported by Rignot et al., 2013, the 1321 Gt/yr reported by Depoorter et al. 2013, or the 1026 Gt/yr "steady state" calving flux reported by Liu et al., 2015. The abstract states that the total pan-Antarctic calving flux is 771.1 Gt/yr, when in fact this number merely represents the total amount of calving that was measured.

(Page 2) The wording of the present manuscript suggests that the entire Antarctic coastline is captured annually in this study, but I could not find any explicit statements about whether small ice shelves or marine-terminating glaciers without ice shelves were included.

(Page 2) Regarding the 1 $km^2$ threshold that must be met for icebergs to be included in estimates of calving flux, the authors point out that although small icebergs are high in number, their contribution to the total calving flux can be neglected because the icebergs are so small. This argument needs evidence.

(Page 4) **Is Antarctica growing in extent?** One possible way to reconcile the 771.1 Gt/yr value reported here with the >1000 Gt/yr calving flux reported in previous studies—and that is if the ice sheet is growing in extent. The static flux gates used in previous studies may indeed measure a flow of >1000 Gt/yr passing by, but if the ice does not calve after passing through the flux gate, then the ice sheet grows, and the methods presented here would capture a lower and more accurate measurement of true calving behavior. Such was suggested by Liu et al., 2015 to reconcile their measurement of 755 Gt/yr with their own estimate of 1026 Gt/yr of "steady state" calving.

I will note that if you add ~150 Gt/yr of calving from marine-terminating glaciers, plus ~100 Gt/yr of icebergs smaller than 1 km2, plus the 271 Gt/yr rate of ice shelf growth due to area extent reported by Liu et al, to the 771 Gt/yr reported here, the result is about 1300 Gt/yr, which falls between the flux-gate calculations of Rignot et al. and Depoorter et al. Is it a coincidence that this perfectly closes the mass budget and reconciles the difference between the two measurement techniques? It would be insightful to read the authors' take on this in the Discussion section.

If the ice sheet is in fact changing in extent, then it seems worth reporting an annual time series of ice sheet area that can be directly compared with the annual time series of calving area. Given that this work involved the development of annual coastlines that purportedly cover the entire continent, it should be trivial to plot a time series of the area enclosed by the coastline each year, and this would provide valuable context for understanding what is or isn't included in the calving flux data.

**The Discussion section** does very little discussing. In its present form, this section repeats several numbers from earlier in the manuscript, but there is no discussion of what the numbers mean or how we should interpret them. There's also no mention why the annual calving fluxes in this paper differ so greatly from those reported by Rignot et al., 2013 or Depoorter et al., 2013.

Response 1:

Thank you for your suggestion.

We stated that we observed the calving rate of Antarctic ice shelves rather than the steady-state iceberg calving flux presented by Rignot et al.2013 and Depoorter et al. 2013. The calving flux is calculated based on a standard budget method, in which they assumed a steady-state calving front for a given set of ice thicknesses and velocities along with the ice front gate. Calving flux is found by integrating ice-shelf thickness and ice velocity along the calving front, it may overestimate iceberg calving where near-ice-front melting is substantial and calving is infrequent; conversely, large icebergs may on average be thicker than the ice front, in which case ice-front fluxes underestimate calving.

As described by our previous work (Liu et al., 2015): "the steady-state iceberg calving, is defined as the calving flux necessary to maintain a steady-state calving front for a given set of ice thicknesses and velocities along with the ice front gate (Rignot et al., 2013; Depoorter et al., 2013). These studies indirectly inferred iceberg calving assuming a steady-state calving front, neglecting the contribution of advance or retreat of the calving front to the mass balance of ice shelves (Rignot et al., 2013; Depoorter et al., 2013). Such "flux gate" calculations are inevitably biased, as they underestimate iceberg calving for retreating ice shelves or overestimate it for advancing ice shelves. This deficiency is problematic not only for estimates of the mass balance of ice shelves but also because current models of iceberg calving provide conflicting predictions about whether increased basal melt will lead to an increase or decrease in iceberg calving.".

In this study, we used an improved calving observation method (Qi et al.,2020) and extended Liu et al. (2015)'s 6-yr iceberg calving observations to 14-yr (now updated to 15-yr) observation. Different from the estimation of calving flux, estimating the mass balance of ice shelves out of steady-state, requires additional information about the change of ice thickness and the change of areal extent of the ice shelf, which is determined by the advance or retreat of the calving front. Observations show that both iceberg calving and ice shelf extent change over the observational period, proving that the steady-state calving front assumption is invalid. In comparison, our dataset more realistically reflects the occurrence of calving at the continental scale.

Here we avoid the assumption of steady-state calving front by combining traditional estimates of ice shelf mass balance with an annual record of iceberg calving events larger than 1 $km^2$ from all Antarctic ice shelves for the period August 2005 to August 2020. The annual calving mass is calculated by sum up the calving mass of each observed single calving event and then divide by 15 to get the annual average value. 770.3 Gt/yr is the 15-yr average calving rate of all the Antarctic ice shelves between 2005 and 2020. Small ice shelves were included, but marine-terminating glaciers were not included (our iceberg calving extraction only included calving from ice shelves but did not include marine-terminating glaciers, and the boundaries of ice shelves are referenced from the MEaSUREs Antarctic Boundaries Version 2 released by NSIDC).

The 1089 Gt/yr value reported by Rignot et al. (2013) and the 1026 Gt/yr "steady-state" calving flux reported by Liu et al. (2015) were the steady-state iceberg calving flux of the Antarctic ice shelves. The 1265 Gt/yr value reported by Rignot et al., (2013) and the 1321 Gt/yr reported by Depoorter et al. (2013) were the steady-state iceberg calving flux around the whole Antarctic coast including marine-terminating glaciers. Based on Rignot et al.'s (2013) estimation, there are 176 Gt/yr of calving from marine-terminating glaciers.

We have extracted all calving events larger than 0.05 $km^2$ from August 2015 to August 2019 to explore the suitable calving detection threshold. In the observed period, 2032 annual Antarctic calving events were detected with areas ranging from 0.05 $km^2$ to 6141.0 $km^2$ (Table S1, Qi et al., 2020). Among them, there were 1209 calving events smaller than 1 $km^2$, with a total area of 483.7 $km^2$ and account for 2.5% of the total calved area. Compared with the 1-10 $km^2$ calving, the <1 $km^2$ calving scale decreases, its frequency increases exponentially, which means that the monitoring workload also increases exponentially, but its calved area

decreases exponentially.

Table S1. Statistics on iceberg calving at different scales from August 2015 to August 2019.

| Year | | <1 km² | 1–10 km² | 10–100 km² | 100–1000 km² | >1000 km² | Total |
|---|---|---|---|---|---|---|---|
| 2015–2016 | Number of calving events | 322 | 162 | 34 | 9 | 1 | 528 |
| | Calved area | 134.0 | 515.4 | 1116.8 | 3058.3 | 893.9 | 5718.4 |
| | Area ratio | 2.3% | 9.0% | 19.5% | 53.5% | 15.6% | - |
| 2016–2017 | Number of calving events | 210 | 167 | 50 | 6 | 1 | 434 |
| | Calved area | 91.2 | 563.0 | 1478.2 | 1077.9 | 6141.0 | 9351.4 |
| | Area ratio | 1.0% | 6.0% | 15.8% | 11.5% | 65.7% | - |
| 2017–2018 | Number of calving events | 361 | 145 | 21 | 2 | 0 | 529 |
| | Calved area | 135.4 | 460.1 | 516.8 | 409.5 | - | 1521.7 |
| | Area ratio | 8.9% | 30.2% | 34.0% | 26.9% | - | - |
| 2018–2019 | Number of calving events | 316 | 198 | 22 | 5 | 0 | 541 |
| | Calved area | 123.1 | 610.1 | 478.3 | 1717.9 | - | 2929.5 |
| | Area ratio | 4.2% | 20.8% | 16.3% | 58.6% | - | - |

Based on the area ratio between the <1 km² and 1-10 km² calving and the 1-10 km² calving rate of 65.6 Gt/yr, we estimated that the <1 km² calving rate of 18.4 Gt/yr. The 1026 Gt/yr "steady-state" calving flux reported by Liu et al. (2015) subtract the <1 km² calving rate of 770.3 Gt/yr and the <1 km² calving rate of 18.4 Gt/yr, the result of 237.3 Gt/yr is the ice shelf growth due to area expansion. Thus, Antarctica is growing in extent.

We also added these discussions in the discussion section: "The trade-off between workload and uncertainty reduction is another consideration in choosing the minimum spatial scale of calving observation. With the calving scale decreasing from 100 km², the number of annual calving events increases exponentially, which means that the monitoring workload also increases exponentially (Qi et al., 2020). Although direct calving observation has the minimum valid extraction area of 0.05 km² based on 75-m SAR resolution images (Qi et al., 2020), it is uneconomical to observe calving events of less than 1 km² using exponentially increasing manual workload to reduce slightly the uncertainty of the total calving-rate estimation. This is why in the present work the calving area and mass of calving events of less than 1 km² of the Antarctic ice shelves were estimated based on observation-area ratio and direct observation of 1–10 km² calving events.

The total circum-Antarctic iceberg calving rate of 955.4±51.4 Gt/yr between 2005 and 2020 observed and estimated in the present study is less than the steady-state iceberg calving fluxes of 1,265 Gt/yr estimated by Rignot et al. (2013) and 1,321 Gt/yr estimated by Depoorter et al. (2013), respectively. The steady-state calving flux is the calving flux necessary to maintain an assumed steady-state calving front for a given set of ice thicknesses and velocities along with the ice-front gate (Rignot et al., 2013; Depoorter et al., 2013). Such "flux-gate" calving calculations for the marine-terminating glaciers are suitable. Our estimated calving rate of the marine-terminating glaciers, 166.7±15.2 Gt/yr, is very close to that reported by Rignot et al. (2013), i.e., 176 Gt/yr. However, such "flux-gate" calving calculations for ice shelves are inevitably biased as they underestimate iceberg calving for retreating ice shelves or overestimate it for advancing ice shelves. Our observed average calving rate of 770.3±29.5 Gt/yr from calving events larger than 1 km2 between 2005 and 2020 is slightly greater than the average rate of 755 Gt/yr between 2005 and 2011 (Liu et al., 2015), which is contributed by two distinct high calving rates of 1,398.8 Gt/yr in 2015/16 and 1,832.6 Gt/yr in 2016/17,

respectively. The average calving rate of 788.7±36.2 Gt/yr of all of the Antarctic ice shelves between 2005 and 2020 is the sum of 770.3±29.5 Gt/yr and the estimated average calving rate of 18.4±6.7 Gt/yr from calving events less than 1 km2, which is less than the steady-state calving fluxes of 1,089 Gt/yr estimated by Rignot et al. (2013) and 1,026 Gt/yr estimated by Liu et al.(2015), respectively. Thus, the Antarctic ice shelves are growing in extent."

**Comment 2:**

(Page 2-4) What we know about the relationship between iceberg size and relative abundance is that they generally follow a Pareto distribution. This is also described as the Gutenberg-Richter relationship. Here, for example, is one figure from the Åström et al. 2014 paper that's cited in the present manuscript…\

In addition to adding some essential caveats to the 771.1 Gt/yr value presented in the abstract, I suggest including a few sentences in the Discussion section about what isn't captured in this dataset. Such a discussion might even exploit the Gutenberg-Richter relationship. For example, below I've assumed a Pareto distribution for the 1786 calving events in this dataset, and by fitting a line in log-log space I'm able to extrapolate this relationship down to a hypothetical iceberg size of 1 m$^2$. This gives us some insight about how many 1 m$^2$ icebergs are likely to calve each year, and how many 10 m$^2$ icebergs, and how many 100 m$^2$, and so on. By knowing the number of icebergs that likely exist, but are too small to be included in this dataset, we can start to build some intuition for how much calving flux might be missing from the dataset.

Response 2:

This is a good point. Thank you for your suggestion. Åström et al. 2014 demonstrated that the probability of calving events obeys a particular pattern no matter if they are small or large events—much like the Gutenberg-Richter law for earthquakes. It was interpreted with a catalog of observations that spans 12 orders of magnitude, with data from Alaska, Svalbard, Greenland, and Antarctica (Fig. 2b from the Åström et al. 2014 paper). The author of this article, Yan Liu, co-authored the Åström et al. 2014 paper and collected, processed or interpreted the calving observations calving volumes range from $10^8$ to $10^{12}$ m$^3$ of Antarctic ice shelves between 2005 and 2011. Effective smaller scale calving observation requires higher resolution remotely sensed imagery.

[Figure]

We also added the related discussion: "The interpretation of calving records spanning 12 orders of magnitude from 1 to $10^{12}$ m$^3$ have demonstrated that the probability of calving events obey a particular pattern whether they are small or large events—much like the Gutenberg-Richter law for earthquakes (Åström et al., 2014). Thus, the fine-scale and continuous observation of calving can be used to investigate how close particular glaciers are to their critical point, and thus how sensitive they may be to near-future changes in

climatic and geometric conditions. However, finer-scale direct observation is greatly limited by the accessibility of high-resolution remotely sensed imagery and significant manual overhead. Our observations provide records of calving volumes ranging from $10^8$ to $10^{12}$ m$^3$ of Antarctic ice shelves."

**Comment 3:**

(Page 5) Section 3.3.3 is relatively innocuous, but it's unclear what value there is in this type of subjective binning. The thresholds were not set by any sort of natural clustering in the size-vs frequency distribution, and no evidence is given that there is any meaningful glaciological distinction between behavior of "low-frequency" and "high-frequency" calving events. Accordingly, the parameters in Table 3 seem somewhat arbitrary, and I'm not sure what can be gained by classifying a calving event as either "low-frequency" or "high-frequency". I recommend either removing this section from the paper or expanding a little bit on why the bins are meaningful and how they might be interpreted in a glaciological or oceanographic context.

Response 3: Thanks for your suggestion. We have removed the classification of "low-frequency" and "high-frequency" calving events for it seems confused to other readers.

**Comment 4:**

Given that this is a data paper, I suggest including a genuine discussion about what insights can be obtained directly from the data or what benefits are gained by any new methods presented here. To be clear, a list of characteristics of the data is not very insightful on its own. Simply stating that Antarctica calves 127.6 icebergs per year does little for readers without any discussion of why that number is important. A sentence that lists the years of elevated calving flux is somewhat meaningless without talking about why calving rates are high some years or whether interannual variability is dominated by a few large calving events or by broadband increases in calving. Likewise, if there is some sort of insight to be gained by knowing that the formal estimate of calved area uncertainty is 17.1 km$^2$, then by all means, discuss that here. Does the area uncertainty represent some limitation of the dataset? Discuss that here. Can the measurement uncertainty be used to offer readers any words of caution about how to interpret the data? Discuss that here. Can we use these 14 years of observations to understand calving processes that occur on multi-decadal timescales? Discuss that here. If there's anything else that feels important to understanding this data, then please discuss it here.

Response 5: Thanks for your suggestion. We have rewritten the discussion part and added the above content.

**Comment 6:**

**Missing coastline shapefiles:** A handful of papers have been published recently, each claiming to have mapped the Antarctic coastline at annual resolution, but to my knowledge, no such data has been made publicly available by any group. Given the aggressive open-data policy of ESSD and my own personal interest in obtaining such a dataset, I feel compelled to ask about the whereabouts of the annual coastlines that were developed to generate this calving flux dataset. Will the annual coastlines be made available?

Response 6: Thanks for your suggestion. As an intermediate product of calved area extraction, we mapped the frontal edge of Antarctic ice shelves at annual resolution. It's different from the coastline product, for it only portrays the position of the **ice-shelf frontal edge** from year to year (as the examples shown in the figure below), and it directly reflects the change in the area of each ice shelf under the assumption of unchanged grounding line position. We will make the annual coastlines of the ice-shelf frontal edge available when the paper is accepted.

[Figure]

**Comment 7:**

**File format:** The dataset is currently made available as a .rar compressed file. That's a somewhat uncommon format, at least in the United States, and it required me to download special software to decompress the file. Users may experience less of a barrier if instead the data are zipped up in an ordinary, open-format .zip file.

Response 7: Thanks for your suggestion. We have made a new version compressed in .zip format. You can get the updated dataset from the National Tibetan Plateau Data Center (http://data.tpdc.ac.cn/en/), and entitled "Annual iceberg calving dataset of the Antarctic ice shelves (2005-2020)" with DOI: 10.11888/Glacio.tpdc.271250

**Comment 8:**

**Polygon type:** I experienced a very minor issue that I could not read the shapefile data in Matlab, because the polygons were saved as PolygonM or PolygonZ format rather than simple Polygon format. To get around this, I had to open each shapefile in QGIS and re-save as Polygon format before I could open them in Matlab. I work with a lot of shapefiles in Matlab, but this is the first time I've encountered this particular issue. I'm not sure if the inability to read PolygonM or PolygonZ format is specific to Matlab, but it may help more people use the data if it's saved as a plain Polygon format.

Response 8:

Thanks for your suggestion. We have made a new version that can be easily reading in Matlab. You can get the updated dataset from the link in Response 7.

```
>> map=shaperead('C:\Users\lenovo\Desktop\Annual_Calving_2005to2020\Annual_Calving_2005to2020_AIS_V2.shp')

map =

1975x1 struct array with fields:
    Geometry
    BoundingBox
    X
    Y
    ID
    YEAR
    Perimtr_KM
    AREA_KM
    SCALE
    THICKNES_M
    VOLUME_KM
    MASS_GT
    RECURRANCE
    UA_KM
    UH_M
    UC_KM
    REGION
    ICESHELF
```

**Comment 9:**

**Named icebergs and known collapse events:** A few well known calving events occurred during the study period, but they can't be directly queried in the shapefile data. It would be helpful if the attributes of the shapefiles contained iceberg names and approximate dates of major events such cases as the Mertz Glacier tongue calving event of 2010, or iceberg A68 at Larsen C in July 2017, or the successive collapse events at Wilkins Ice Shelf.

Response 9:

Thanks for your suggestion. We have made a new version with the name of the specific ice shelves where each calving occurred. The annual calving events are different from the calved icebergs, for it may consist of many single calving happened in the same year (see as the figure below), therefore we could not match the name of the iceberg with every annual calving events.

[Figure]

**Comment 10:**

   **Line 28** is problematic as it currently reads, "In total, 1786 annual calving events occurred on the Antarctic ice shelves from August 2005 to August 2019." The wording of this sentence could easily be misinterpreted, because it says without

   qualification that the *total* number of calving events during the study period was just 1786. No doubt this is an underestimate, likely by an order of magnitude or more. For example, the previous paper by Qi et al. reports that 2032 calving events were detected just within the final four years of the present study's period of investigation. I suggest a very simple fix, which is to say something like "we detect a total of 1786 calving events larger than 1 km2 ..."

   Response 10: Thanks for your suggestion. We have made the correction.

**Comment 11:**

   **Line 187-192:** Calving area uncertainty is also influenced by velocity uncertainty, and should probably be accounted for here.

   Response 11:

   Thanks for your question. We have quantitatively described the error introduced by the velocity in the paper before (see Qi et al., 2020)., in which the advantages and errors of the calving event extraction method are systematically demonstrated. The estimation of calving area uncertainty has already considered the influence of velocity uncertainty. In the calving area detection, we had a step to move back the detected calving

area by simulated coast to its location before calving and modified the calving boundary difference due to velocity uncertainty. Then the calving area uncertainty is dependent on its length of the boundary. i.e., the perimeter.

**Comment 12:**

**Table 4** took me a while to understand, mainly because I was thrown off by the unitless use of the word *frequency*. The usage of the word frequency is not incorrect, per se, but it is slightly more difficult to parse than simply discussing the number of events that were counted. I think the title should be something like "Number of calving events detected in MODIS and SAR." Likewise, the "Calving frequency" row should be renamed "Number of calving events". It's not immediately clear what is meant by the row labeled "Calving area". Does it mean "Total calved area"? The Scale column contains only text descriptors of Area categories that were binned

based on somewhat arbitrary thresholds, and to interpret them in this table the reader is tasked with going back to Table 3 to figure out what these categories mean. I recommend removing the subjective labels like "Medium-scale" and "Extra-large-scale" throughout the paper and replacing with the area range.

Response 12: Thanks for your suggestion. We have made the correction.

Table 4: Frequency and area distribution of different scale calving events derived from MODIS and SAR for 2016/17

| | Scale | MODIS | SAR | Δ(MODIS-SAR) | Δ(MODIS-SAR)/$SAR_{Total}$ |
|---|---|---|---|---|---|
| Number of calving events | Small-scale (< 10 km$^2$) | 163 | 167 | -4 | -1.8% |
| | Medium-scale (10-100 km$^2$) | 50 | 50 | 1 | 0.4% |
| | Large-scale (100-1,000 km$^2$) | 6 | 6 | 0 | - |
| | Extra-large-scale (>1,000 km$^2$) | 1 | 1 | 0 | - |
| | Total | 220 | 224 | -4 | -1.8% |
| Total calved area (km$^2$) | Small-scale (< 10 km$^2$) | 511.0 | 563.0 | -52.0 | -0.6% |
| | Medium-scale (10-100 km$^2$) | 1,441.0 | 1,478.2 | -37.2 | -0.4% |
| | Large-scale (100-1,000 km$^2$) | 1,057.9 | 1,077.9 | -20.0 | -0.2% |
| | Extra-large-scale (>1,000 km$^2$) | 6,054.7 | 6,141.0 | -86.3 | -0.9% |
| | Total | 9,064.6 | 9,260.2 | -195.5 | -2.1% |
| Standard deviation of total calved area (km$^2$) | Small-scale (< 10 km$^2$) | 2.3 | 2.2 | 0.1 | 0.0 |
| | Medium-scale (10-100 km$^2$) | 21.3 | 17.9 | 3.4 | 0.2 |
| | Large-scale (100-1,000 km$^2$) | 93.4 | 91.9 | 1.5 | 0.0 |
| | Extra-large-scale (>1,000 km$^2$) | 0.0 | 0.0 | 0.0 | - |
| | Total | 397.2 | 402.8 | -5.6 | -1.4% |

**Comment 13:**

**Section 5.1** contains several uses of the phrase *calving frequency* to mean the total number of calving events that were detected. The distinction may just be a minor wording preference of mine, but it feels significant as the *calving frequency* implies an intrinsic property of the system, whereas the *number of detected calving events* is unambiguous and directly describes what has been measured. For clarity, I recommend replacing *calving frequency* throughout the paper. This would also make room for another phrase used in this paper, *calving recurrence interval*, which might be easily confused with *calving frequency* if both phrases are used in the same manuscript.

Response 13: Thanks for your suggestion. We have made the correction.

**Comment 14:**

**Figure 5** is interesting, but for I would change the phrase "calving frequency" to "number of calving events", change the phrase "calving area" to "total calved area", remove the entire right-hand column, as the colored bins of area shown in the left-hand plots are a natural proxy for frequency.

Response 14: Thanks for your suggestion. We have made the correction.

[Figure]

**Figure 5: Temporal distribution of annual calving events at different scales of Antarctic ice shelves from August 2005 to August 2020. Panels (a), (b), and (c) present the annual number of calving events, calved area, and calved mass at four scales, respectively. Horizontal dashed lines in Panel (c) denote the 1026 Gt/yr "steady-state" calving flux of ice shelves reported by Liu et al. (2015)**

**Comment 15:**

**Section 5.3** dives into the recurrence interval data, but the concept of the recurrence interval hasn't been adequately described. A fair attempt was made to give sort of a dictionary definition of the term on Line 215, but then by Line 216 the meaning gets lost in vague terms about calving that occurs in the same neighborhood, which depends on some threshold of distinguishing between in-neighborhood and out-of-neighborhood events, and by now I've lost any sense of what the recurrence interval can tell us.

After Section 3.3.3 in which the quantification method is described, the topic of recurrence intervals is comes up again in Section 5.3 without any conceptual bridge to help readers understand physically what is meant by this sentence that begins on Line 307:

*Calving events with a recurrence interval of 3 had the highest frequency, ...*

Already I'm confused, as I have no physical intuition for what the recurrence interval tells us about glaciers, or if this is just a characteristic of the detection limits of the method. Without setting the stage for understanding what the recurrence interval actually tells us, readers are likely to left wondering how once-every-three-year calving events can have a higher frequency than annual calving events. After hitting this confluence of conceptual roadblocks so early in the sentence, it's difficult then to understand the remainder of the sentence, which continues,

*...accounting for 18.8% of the total and occurring 335 times in 14 years, followed by those with recurrence intervals of 5, 2 and 4, accounting for 18.5%, 15.2% and 14.3%, respectively.*

Again, what do any of these numbers mean? My interpretation is that the methods presented in this paper may not fully capture the types of calving events that occur more often than every three years, because the high-frequency events are more likely to be smaller and go undetected. If that's the case, that's absolutely fine, but explore the concept further, so the folks who use this data will understand what it tells us and what the limitations are. If, on the other hand, there's something glaciologically meaningful here, then please help readers understand it.

…

**Figure 6** is difficult to interpret. I know it's supposed to be communicating something, but I keep gazing into the figure, hoping it will reveal its secrets to me. After some inspection I see that area and mass well correlated, but that is to be expected. And it appears that high numbers of the small icebergs combine to represent a fair portion of the total detected mass, but still

I'm not sure what the take-home message of this figure is.

I think the units of the horizontal axis are supposed to be years, but there are no data in the 1-year bin, so I must wonder,

Are there truly *no* places in Antarctica where calving occurs every year, or is this just a limitation of the detection method? Or am I misunderstanding the meaning of recurrence interval?

I also suspect there's some intention behind the shading of the left and right side of the figure, but I can't figure out what it's trying to communicate.

The task of interpreting this figure is not made any easier by the caption, which contains only a verbless sentence fragment. From the main body of this manuscript, it is clear that the authors are excellent writers, so by all means, use your talents to fill this space with vivid descriptions of what's happening in the figure! Help me understand what I should be noticing in the graphic, and help me understand why it's important.

Response 15:

Thank you for your comments. We modified the description of calving recurrence interval.

Section 3.4.3: "Most calving events are thought to be part of the natural cycle of advance and retreat of ice shelves. Calving recurrence means that a calving event with the same spatial scale reoccurs at the same calving front. The recurrence interval of a calving event is defined as the year interval between the two recurrence calving events."

Section 5.4: "The recurrence interval of calving provides additional qualitative information about the style of calving (Liu et al., 2015) and determines the suitable observation period for identifying ice shelf nonsteady-state behavior. For example, the rift-opening calving of the Amery Ice Shelf has reoccurred in 2019 since the last calving in 1963/64 (Li et al., 2020), detach along the boundary of isolated pre-existing rifts for decades. The observational records spanning many decades would be needed to determine its nonsteady-state behavior. In contrast, more frequent disintegration calving events are mainly caused by the hard to observe rapid basal crevasse propagation (Liu et al., 2015). The calving front retreat associated with these frequent calving events can be robustly identified over a short observation period due to the shorter recurrence intervals. In other words, the calving events with shorter recurrence intervals are more sensitive to current climate change.

Figure 6 (a) shows the calving recurrence interval is little related to calving scales of caving.  The two extra-large-scale (> 1,000 km$^2$) calving events reoccurred on the Thwaite Glacier during our observed period indicating its distinct retreat, while the other four extra-large-scale events from the Larsen C, Wilkins, Totten,

and Amery Ice shelves did not reoccur. Figure 6 (b) shows that 76% of the total number of calving events reoccurred during the observed period (i.e., their recurrence intervals of calving are less than 8 yr), which suggests that the annual calving number is likely to be an indicator of the response of calving to climate change. Nearly half of the cumulative calved area from the events with the recurrence intervals greater than 8 yr (i.e., the events only occurred once during the observed period) suggests that the annual calved area is not suitable for identifying the nonsteady-state behaviors of some ice shelves."

We modified Figure 6, see below:

[Figure]

**Figure 6: Distribution of calving events with different recurrence intervals. Panel (a) shows the cumulative number of calving events at different scales. Panel (b) shows the cumulative percentages of the cumulative number of calving events, the cumulative calved area, and the cumulative calved mass.**

**Comment 18:**

**Line 344** says "the Totten Ice Shelf was collapsing every year." The word "collapsing" may be a bit too strong, so consider changing to something like "We detect calving events at Totten Ice Shelf every year."

Response 18: Corrected.

**Comment 19:**

**Lines 350 to 363** rehash numbers from earlier in the manuscript without reframing or adding any new perspectives. I think this paragraph can be deleted without any detriment to the paper.

Response 19: Thanks for your suggestion. We have made the correction and rewritten this section.

**Comment 20:**

ENSO analysis seems out of place: The attribution of calving events to ENSO anomalies and surface melting looks very interesting, but the analysis feels out of place in this data paper, particularly as the topic is only introduced in the final closing remarks of the paper. If these scientific results are compelling as they appear to be, then they should be described in detail in a separate paper, where they can be discussed at length while being given a chance for proper peer review.

**Lines 363 to 373** convey a level of enthusiasm that should absolutely be harnessed to develop this analysis further. The early results look very interesting indeed, but I don't think the Discussion section of a data paper is the appropriate place to present new scientific findings about correlations between ENSO and

iceberg calving. I recommend removing these two paragraphs and Figure 9 from the paper.

Response 20:

Thanks for your suggestion but we prefer to keep this part.

Our original intention of producing a long time series of fine observation datasets was to analyze the characteristics of calving in the background of climate change. The most important advantage of our annual observed calving rather than the steady-state calving flux is that it can reflect the calving response to climate change. The correlations between ENSO and iceberg calving demonstrate this point. It also demonstrates that the dataset can be used to investigate the external atmospheric and oceanic impacts on iceberg calving. This is a data description paper, we will concentrate more on the description of the data itself. In the discussion section we show a preliminary application of this data, that is, the attribution of calving events to ENSO anomalies and surface melting. We are also conducting in-depth research and hope that will be given a chance for peer review in the upcoming manuscript.

---

## Author Comment (AC3)

**Response to Anonymous Referee #2**

We sincerely thank the reviewer for the insightful and constructive comments and important suggestions. These comments were very helpful for revising and improving our paper. We have endeavored to address all of the comments and to improve the manuscript to the best of our ability. Our point-by-point responses are detailed below.

**Comment 1: P2, L55, the authors grouped the state of arts by different spatial resolution, not by the detection method?**

Response 1:

Thanks for the comment. In P2, L55, we make a short literature review of the existed calving monitor results. The detection methods of those studies can be classified into three categories, automatic extraction, semi-automatic extraction, and manual extraction. More detailed information on different calving detection methods can be seen in (Qi et al., 2020), in which we introduced the extraction method we used during producing this dataset and also summarized other calving extracting methods. In this manuscript, we grouped the state of arts by different spatial resolution in order to make a comparison between the existed calving monitor results through the dimension of spatial and temporal resolution. Then, we found the gap that there is a lack in the long time series, high spatial resolution, and wide coverage calving dataset. That's why we made such a dataset.

**Comment 2: P6, Section 3.2, it's not clear here why August 2005, 2010 and 2015 were used as the input benchmark. According to the descriptions in this section, every year's actual coastline modified from last year's extraction was used as the input of next year.**

Response 2:

Thanks for the comment and it's a really good question. Theoretically, every year's actual coastline modified from last year's extraction was used as the input of next year, so we can use only one benchmark coastline to generate the next-yr coastline and iterate this step to get the coastlines of the following years. However, while modifying the simulated coastline derived from the last-yr coastline and ice velocity, we mainly modify the coastline near the calved area to make sure it fit in the actual coastline from the satellite imagery and manually correct the system errors at the regional scale. That means the rest of the simulated coastline where calving did not occur may have systematic errors (introduced by ice velocity product). Here, in order to reduce the error caused by the ice velocity during the iterative calculation of the coastline, we divided 14 consecutive years into three intervals. We used coastlines of 2005, 2010, and 2015 (checked and manually corrected) as the benchmark to control the error of coastline simulation within five years.

**Comment 3: P8, L178, to my knowledge, Bedmap 2 may not provide thickness data in some of the coast areas, how do you deal with such situation.**

Response 3:

Thanks for your comment. During the production of the dataset, we also considered this situation. We tried to solve the problem in the following three steps. First, we masked out the ice-shelf zone thickness in Bedmap 2. Second, we extracted the average thickness of each calving event from the masked ice thickness through step 1. Then, we checked the average thickness of all calving events. For missed and abnormal values (results show that they only account for a small proportion of the total), we moved the polygon backward

along the ice flow to the calving front where there is thickness data coverage. After that, we re-extracted the average thickness of those calving events to make sure they are given appropriate thickness. Besides, we also used Bedmachine as a supplement and validation.

To reduce misunderstandings among readers, we have added the following description to the manuscript.

**3.4.1 Calved area and calved mass**

After acquiring vectors of the calved area polygons, we calculated their areas under polar projection. Then, these values were divided into four different scales: small-scale (1-10 km2), medium-scale (10-100 km2), large-scale (100-1,000 km2), and extra-large-scale values (>1,000 km2). We further obtained the average thickness of each calved area from the Antarctic ice thickness products (Bedmap 2 and Bedmachine). First, we masked out the ice-shelf zone thickness in Bedmap 2 and Bedmachine. Second, we extracted the average thickness of each calving event from the masked ice thickness through step 1. Then, we checked the average thickness of all calving events. For missed and abnormal values (results show that they only account for a small proportion of the total), we moved the polygon backward along the ice flow to the calving front where there is thickness data coverage. After that, we re-extracted the average thickness of those calving events to make sure they are given appropriate thickness.

**Comment 4: P10, Section 3.3.3, this section is not clear, how to define the center point, the perimeter of a calving area? Line 224-L228, this paragraph is confusing.**

Response 4:

Thanks for your question. Each calving event was recorded as a polygon depicting its boundaries. We used the function "Feature to Point" in ArcMap to get the center points of each individual calving polygon. For an input polygon feature, the location of the output point will be determined as the center of gravity (centroid) of the polygon. As for the perimeter of a calving area, we calculated it through the function "Calculate Geometry" in ArcMap. To reduce the confusion about this paragraph, we rewrote section 3.3.3 (now section 3.4.3 in the manuscript) and made the following modifications.

**3.4.3 Recurrence interval**

Calving recurrence means that a calving event with the same spatial scale reoccurs at the same calving front (Liu et al., 2015), which are usually thought to be part of the natural cycle of advance and retreat of ice shelves. The recurrence interval of a calving event, a measurement of the natural calving cycle, is defined as the year interval between the two recurrence calving events. To acquire this attribute, we performed the following work. First, we get the perimeter of each calving polygon through the function "Calculate Geometry" in ArcMap. Based on that, we calculated the average perimeter of all calving events at the same scale for 15 years. We defined the Buffer radii as half of the average perimeters at different scales rounded upwards to the nearest integer. The specific values used for this dataset are shown in Table 3.

…

After that, we used the function "Feature to Point" in ArcMap to get the center points of each individual calving polygon. For an input polygon, the location of the output point will be determined as its center of gravity. Then, we build buffers for each calving center point based on the radii calculated in the previous steps. For each calving event, we count the number of calving center points with the same scale

that falls into its buffer. For buffers that fall into more than two points, the calving recurrence interval is defined as the total number of years (15) divided by the exact number of calving center points falling within. For buffers with only one point, the calving recurrence interval is defined as the greater value of time intervals between these calving events and boundary years (2005 or 2020).

**Comment 5: P10, L230, the iceberg calving events were divided into two types, high frequency and low frequency; it seems contradictory with calving frequency (Table 4). Is calving frequency means the number of calving events in every year?**

Response 5:

Calving frequency in Table 4 represents "the number of calving events". While the definition of high-frequency calving and low-frequency calving is classified based on its calving recurrence interval.

In the beginning, we divided calving events according to the level of intensity of their occurrence and tried to categorically explore their response to climatic and environmental factors. According to the calving recurrence interval, we classified the annual iceberg calving events into two different types: high-frequency (calving recurrence interval of ≤7 years) and low-frequency calving events (calving recurrence interval of >7years). The longer the recurrence interval is, the less calving that occurs in a given period and the lower the frequency.

After careful consideration, we decided that this classification criterion might be confused and it is not the main nature of this dataset, so we removed it. Also, we have replaced the expression "calving frequency" with "number of calving events" for better understanding

**Comment 6: P11, Table 4, please also include the standard deviation for the calving areas.**

Response 6: Thank you for your suggestion. We have added the standard deviation for the calving areas in Table 4.

**Comment 7: P12, add standard deviation in section 4.2.**

Response 7: Thank you for your suggestion. We added the standard deviations of area uncertainty, thickness uncertainty, and mass uncertainty.

**Comment 8: The calving events with high uncertainty should be excluded or discussed separately in Section 5.**

Response 8: Thank you for your suggestion.

The uncertainty is calculated based on the equations in section 3. The calving events with high uncertainty don't mean their existence is not justified. The uncertainty does not represent the reliability of the calving extraction results, but the reliability of attribute calculation results. For example, the calved area uncertainty is mainly determined by the perimeter of each single calving event. In the case of the same area, a long and narrow calving area has higher uncertainty than a square calving area. Thickness uncertainty is mainly attributed to firn depth correction. The calved mass uncertainty is mainly determined by thickness uncertainty.

In order to ensure the accuracy of calving extraction, we performed manual checking, semi-automatic extraction, and removal of calving events with an area less than 1 km$^2$. Therefore, we think that the calved area polygons retained in the current dataset are true and reliable.

We also discussed it in Section 6 Discussion. "The calved-area uncertainty of our direct observation (Qi

et al., 2020) is dependent on the spatial resolution of the imagery, uncertainty of velocity data, and the perimeter-to-area ratio of the calved area. In the case of the same area, a long and narrow calving area has higher uncertainty than a square calving area. The relatively low-spatial-resolution satellite imagery used in this work and the characteristic of a long and narrow calving area is the main reasons this method is not suitable for high-accuracy calving observation of marine-terminating glaciers."

---

## Author Comment (AC4)

College of Global Change and Earth System Science

Beijing Normal University

No. 19, Xinjiekouwai Street, HaiDian District

Beijing 100875, China

19/4/2021

Dear Editors,

**Revision of ESSD manuscript ESSD-2020-340-CC1:**

**A 15-yr Circum-Antarctic Iceberg Calving Dataset Derived from Continuous Satellite Observations**

It was a pleasure for us to read the encouraging and constructive comments and important suggestions provided by the reviewers. They were of great assistance in improving the quality of the manuscript. All of the comments and suggestions were considered during the revision process. Moreover, we have provided a new version of the manuscript along with a point-by-point response to all of the reviewers' comments.

The following pages provide the point-by-point responses to the suggestions made by the editor and reviewers and a detailed description of the changes made.

Yours sincerely,

The authors

**Response to Professor Mohammed Shokr**

**Comment 1: Back to the manuscript; it is well written; and the figures are well done. Figure 8 in particular is excellent. The authors put the satellite data, especially the European SAR systems, which are made available for free, to good use. I have no comments regarding corrections except the "15 year" in the Abstract (should be 14).**

Response 1:

On behalf of the manuscript's authors, I'd like to thank Professor Mohammed Shokr for the encouragement and recognition of our work as well as valuable comments to improve our manuscript.

In the original version of the manuscript, line 25 in the abstract shows that "we developed this product based on 15 years of continuous multisource optical and synthetic aperture radar images". Here the number 15 means that we used 15-yr of remotely sensed imagery (from 2005 to 2019), it represents the endpoint of the time interval of the data. While the number 14, as the title suggests, represents the period of the calving product. In other words, we used 15 years of continuous satellite observations to derive a 14-yr circum-Antarctic iceberg calving dataset.

Now, we have extended our dataset from 14-yr to a 15-yr circum-Antarctic annual iceberg calving product. Therefore, we revised the manuscript to avoid misunderstandings in the abstract: "In this study, a 15-yr annual iceberg-calving product measuring every independent calving event larger than 1 $km^2$ over all of the Antarctic ice shelves that occurred from August 2005 to August 2020 was developed based on 16 years of continuous satellite observations."

**Response to Anonymous Referee #2**

**Comment 1: P2, L55, the authors grouped the state of arts by different spatial resolution, not by the detection method?**

Response 1:

Thanks for the comment. In P2, L55, we make a short literature review of the existed calving monitor results. The detection methods of those studies can be classified into three categories, automatic extraction, semi-automatic extraction, and manual extraction. More detailed information on different calving detection methods can be seen in (Qi et al., 2020), in which we introduced the extraction method we used during producing this dataset and also summarized other calving extracting methods. In this manuscript, we grouped the state of arts by different spatial resolution in order to make a comparison between the existed calving monitor results through the dimension of spatial and temporal resolution. Then, we found the gap that there is a lack in the long time series, high spatial resolution, and wide coverage calving dataset. That's why we made such a dataset.

**Comment 2: P6, Section 3.2, it's not clear here why August 2005, 2010 and 2015 were used as the input benchmark. According to the descriptions in this section, every year's actual coastline modified from last year's extraction was used as the input of next year.**

Response 2:

Thanks for the comment and it's a really good question. Theoretically, every year's actual coastline modified from last year's extraction was used as the input of next year, so we can use only one benchmark coastline to generate the next-yr coastline and iterate this step to get the coastlines of the following years. However, while modifying the simulated coastline derived from the last-yr coastline and ice velocity, we mainly modify the coastline near the calved area to make sure it fit in the actual coastline from the satellite imagery and manually correct the system errors at the regional scale. That means the rest of the simulated coastline where calving did not occur may have systematic errors (introduced by ice velocity product). Here, in order to reduce the error caused by the ice velocity during the iterative calculation of the coastline, we divided 14 consecutive years into three intervals. We used coastlines of 2005, 2010, and 2015 (checked and manually corrected) as the benchmark to control the error of coastline simulation within five years.

**Comment 3: P8, L 178, to my knowledge, Bedmap 2 may not provide thickness data in some of the coast areas, how do you deal with such situation.**

Response 3:

Thanks for your comment. During the production of the dataset, we also considered this situation. We tried to solve the problem in the following three steps. First, we masked out the ice-shelf zone thickness in Bedmap 2. Second, we extracted the average thickness of each calving event from the masked ice thickness through step 1. Then, we checked the average thickness of all calving events. For missed and abnormal values (results show that they only account for a small proportion of the total), we moved the polygon backward along the ice flow to the calving front where there is thickness data coverage. After that, we re-extracted the average thickness of those calving events to make sure they are given appropriate thickness. Besides, we also used Bedmachine as a supplement and validation.

To reduce misunderstandings among readers, we have added the following description to the manuscript.

3.4.1 Calved area and calved mass

After acquiring vectors of the calved area polygons, we calculated their areas under polar projection. Then, these values were divided into four different scales: small-scale (1-10 km2), medium-scale (10-100 km2), large-scale (100-1,000 km2), and extra-large-scale values (>1,000 km2). We further obtained the average thickness of each calved area from the Antarctic ice thickness products (Bedmap 2 and Bedmachine). First, we masked out the ice-shelf zone thickness in Bedmap 2 and Bedmachine. Second, we extracted the average thickness of each calving event from the masked ice thickness through step 1. Then, we checked the average thickness of all calving events. For missed and abnormal values (results show that they only account for a small proportion of the total), we moved the polygon backward along the ice flow to the calving front where there is thickness data coverage. After that, we re-extracted the average thickness of those calving events to make sure they are given appropriate thickness.

**Comment 4: P10, Section 3.3.3, this section is not clear, how to define the center point, the perimeter of a calving area? Line 224-L228, this paragraph is confusing.**

Response 4:

Thanks for your question. Each calving event was recorded as a polygon depicting its boundaries. We used the function "Feature to Point" in ArcMap to get the center points of each individual calving polygon. For an input polygon feature, the location of the output point will be determined as the center of gravity (centroid) of the polygon. As for the perimeter of a calving area, we calculated it through the function "Calculate Geometry" in ArcMap. To reduce the confusion about this paragraph, we rewrote section 3.3.3 (now section 3.4.3 in the manuscript) and made the following modifications.

**3.4.3 Recurrence interval**

Calving recurrence means that a calving event with the same spatial scale reoccurs at the same calving front (Liu et al., 2015), which are usually thought to be part of the natural cycle of advance and retreat of ice shelves. The recurrence interval of a calving event, a measurement of the natural calving cycle, is defined as the year interval between the two recurrence calving events. To acquire this attribute, we performed the following work. First, we get the perimeter of each calving polygon through the function "Calculate Geometry" in ArcMap. Based on that, we calculated the average perimeter of all calving events at the same scale for 15 years. We defined the Buffer radii as half of the average perimeters at different scales rounded upwards to the nearest integer. The specific values used for this dataset are shown in Table 3.

…

After that, we used the function "Feature to Point" in ArcMap to get the center points of each individual calving polygon. For an input polygon, the location of the output point will be determined as its center of gravity. Then, we build buffers for each calving center point based on the radii calculated in the previous steps. For each calving event, we count the number of calving center points with the same scale that falls into its buffer. For buffers that fall into more than two points, the calving recurrence interval is defined as the total number of years (15) divided by the exact number of calving center points falling within. For buffers with only one point, the calving recurrence interval is defined as the greater value of

time intervals between these calving events and boundary years (2005 or 2020).

**Comment 5: P10, L230, the iceberg calving events were divided into two types, high frequency and low frequency; it seems contradictory with calving frequency (Table 4). Is calving frequency means the number of calving events in every year?**

Response 5:

Calving frequency in Table 4 represents "the number of calving events". While the definition of high-frequency calving and low-frequency calving is classified based on its calving recurrence interval.

In the beginning, we divided calving events according to the level of intensity of their occurrence and tried to categorically explore their response to climatic and environmental factors. According to the calving recurrence interval, we classified the annual iceberg calving events into two different types: high-frequency (calving recurrence interval of ≤7 years) and low-frequency calving events (calving recurrence interval of >7years). The longer the recurrence interval is, the less calving that occurs in a given period and the lower the frequency.

After careful consideration, we decided that this classification criterion might be confused and it is not the main nature of this dataset, so we removed it. Also, we have replaced the expression "calving frequency" with "number of calving events" for better understanding

**Comment 6: P11, Table 4, please also include the standard deviation for the calving areas.**

Response 6: Thank you for your suggestion. We have added the standard deviation for the calving areas in Table 4.

**Comment 7: P12, add standard deviation in section 4.2.**

Response 7: Thank you for your suggestion. We added the standard deviations of area uncertainty, thickness uncertainty, and mass uncertainty.

**Comment 8: The calving events with high uncertainty should be excluded or discussed separately in Section 5.**

Response 8: Thank you for your suggestion.

The uncertainty is calculated based on the equations in section 3. The calving events with high uncertainty don't mean their existence is not justified. The uncertainty does not represent the reliability of the calving extraction results, but the reliability of attribute calculation results. For example, the calved area uncertainty is mainly determined by the perimeter of each single calving event. In the case of the same area, a long and narrow calving area has higher uncertainty than a square calving area. Thickness uncertainty is mainly attributed to firn depth correction. The calved mass uncertainty is mainly determined by thickness uncertainty.

In order to ensure the accuracy of calving extraction, we performed manual checking, semi-automatic extraction, and removal of calving events with an area less than 1 km$^2$. Therefore, we think that the calved area polygons retained in the current dataset are true and reliable.

We also discussed it in Section 6 Discussion. "The calved-area uncertainty of our direct observation (Qi et al., 2020) is dependent on the spatial resolution of the imagery, uncertainty of velocity data, and the perimeter-to-area ratio of the calved area. In the case of the same area, a long and narrow calving area has higher uncertainty than a square calving area. The relatively low-spatial-resolution satellite imagery used in

this work and the characteristic of a long and narrow calving area is the main reasons this method is not suitable for high-accuracy calving observation of marine-terminating glaciers."

**Response to Professor Chad A. Greene:**

**Comment 1:**

(Page 1) **Under reported value of calving flux**: The 771.1 Gt/yr value of calving flux presented here is significantly lower than the 1265 Gt/yr value reported by Rignot et al., 2013, the 1321 Gt/yr reported by Depoorter et al. 2013, or the 1026 Gt/yr "steady state" calving flux reported by Liu et al., 2015. The abstract states that the total pan-Antarctic calving flux is 771.1 Gt/yr, when in fact this number merely represents the total amount of calving that was measured.

(Page 2) The wording of the present manuscript suggests that the entire Antarctic coastline is captured annually in this study, but I could not find any explicit statements about whether small ice shelves or marine-terminating glaciers without ice shelves were included.

(Page 2) Regarding the 1 km$^2$ threshold that must be met for icebergs to be included in estimates of calving flux, the authors point out that although small icebergs are high in number, their contribution to the total calving flux can be neglected because the icebergs are so small. This argument needs evidence.

(Page 4) **Is Antarctica growing in extent?** One possible way to reconcile the 771.1 Gt/yr value reported here with the >1000 Gt/yr calving flux reported in previous studies—and that is if the ice sheet is growing in extent. The static flux gates used in previous studies may indeed measure a flow of >1000 Gt/yr passing by, but if the ice does not calve after passing through the flux gate, then the ice sheet grows, and the methods presented here would capture a lower and more accurate measurement of true calving behavior. Such was suggested by Liu et al., 2015 to reconcile their measurement of 755 Gt/yr with their own estimate of 1026 Gt/yr of "steady state" calving.

I will note that if you add ~150 Gt/yr of calving from marine-terminating glaciers, plus ~100 Gt/yr of icebergs smaller than 1 km2, plus the 271 Gt/yr rate of ice shelf growth due to area extent reported by Liu et al, to the 771 Gt/yr reported here, the result is about 1300 Gt/yr, which falls between the flux-gate calculations of Rignot et al. and Depoorter et al. Is it a coincidence that this perfectly closes the mass budget and reconciles the difference between the two measurement techniques? It would be insightful to read the authors' take on this in the Discussion section.

If the ice sheet is in fact changing in extent, then it seems worth reporting an annual time series of ice sheet area that can be directly compared with the annual time series of calving area. Given that this work involved the development of annual coastlines that purportedly cover the entire continent, it should be trivial to plot a time series of the area enclosed by the coastline each year, and this would provide valuable context for understanding what is or isn't included in the calving flux data.

**The Discussion section** does very little discussing. In its present form, this section repeats several numbers from earlier in the manuscript, but there is no discussion of what the numbers mean or how we should interpret them. There's also no mention why the annual calving fluxes in this paper differ so greatly from those reported by Rignot et al., 2013 or Depoorter et al., 2013.

Response 1:

Thank you for your suggestion.

We stated that we observed the calving rate of Antarctic ice shelves rather than the steady-state iceberg calving flux presented by Rignot et al.2013 and Depoorter et al. 2013. The calving flux is calculated based on a standard budget method, in which they assumed a steady-state calving front for a given set of ice thicknesses and velocities along with the ice front gate. Calving flux is found by integrating ice-shelf thickness and ice velocity along the calving front, it may overestimate iceberg calving where near-ice-front melting is substantial

and calving is infrequent; conversely, large icebergs may on average be thicker than the ice front, in which case ice-front fluxes underestimate calving.

As described by our previous work (Liu et al., 2015): "the steady-state iceberg calving, is defined as the calving flux necessary to maintain a steady-state calving front for a given set of ice thicknesses and velocities along with the ice front gate (Rignot et al., 2013; Depoorter et al., 2013). These studies indirectly inferred iceberg calving assuming a steady-state calving front, neglecting the contribution of advance or retreat of the calving front to the mass balance of ice shelves (Rignot et al., 2013; Depoorter et al., 2013). Such "flux gate" calculations are inevitably biased, as they underestimate iceberg calving for retreating ice shelves or overestimate it for advancing ice shelves. This deficiency is problematic not only for estimates of the mass balance of ice shelves but also because current models of iceberg calving provide conflicting predictions about whether increased basal melt will lead to an increase or decrease in iceberg calving.".

In this study, we used an improved calving observation method (Qi et al.,2020) and extended Liu et al. (2015)'s 6-yr iceberg calving observations to 14-yr (now updated to 15-yr) observation. Different from the estimation of calving flux, estimating the mass balance of ice shelves out of steady-state, requires additional information about the change of ice thickness and the change of areal extent of the ice shelf, which is determined by the advance or retreat of the calving front. Observations show that both iceberg calving and ice shelf extent change over the observational period, proving that the steady-state calving front assumption is invalid. In comparison, our dataset more realistically reflects the occurrence of calving at the continental scale.

Here we avoid the assumption of steady-state calving front by combining traditional estimates of ice shelf mass balance with an annual record of iceberg calving events larger than 1 km$^2$ from all Antarctic ice shelves for the period August 2005 to August 2020. The annual calving mass is calculated by sum up the calving mass of each observed single calving event and then divide by 15 to get the annual average value. 770.3 Gt/yr is the 15-yr average calving rate of all the Antarctic ice shelves between 2005 and 2020. Small ice shelves were included, but marine-terminating glaciers were not included (our iceberg calving extraction only included calving from ice shelves but did not include marine-terminating glaciers, and the boundaries of ice shelves are referenced from the MEaSUREs Antarctic Boundaries Version 2 released by NSIDC).

The 1089 Gt/yr value reported by Rignot et al. (2013) and the 1026 Gt/yr "steady-state" calving flux reported by Liu et al. (2015) were the steady-state iceberg calving flux of the Antarctic ice shelves. The 1265 Gt/yr value reported by Rignot et al., (2013) and the 1321 Gt/yr reported by Depoorter et al. (2013) were the steady-state iceberg calving flux around the whole Antarctic coast including marine-terminating glaciers. Based on Rignot et al.'s (2013) estimation, there are 176 Gt/yr of calving from marine-terminating glaciers.

We have extracted all calving events larger than 0.05 km$^2$ from August 2015 to August 2019 to explore the suitable calving detection threshold. In the observed period, 2032 annual Antarctic calving events were detected with areas ranging from 0.05 km$^2$ to 6141.0 km$^2$ (Table S1, Qi et al., 2020). Among them, there were 1209 calving events smaller than 1 km$^2$, with a total area of 483.7 km$^2$ and account for 2.5% of the total calved area. Compared with the 1-10 km$^2$ calving, the <1 km$^2$ calving scale decreases, its frequency increases exponentially, which means that the monitoring workload also increases exponentially, but its calved area decreases exponentially.

Table S1. Statistics on iceberg calving at different scales from August 2015 to August 2019.

| Year | | <1 km$^2$ | 1–10 km$^2$ | 10–100 km$^2$ | 100–1000 km$^2$ | >1000 km$^2$ | Total |
|---|---|---|---|---|---|---|---|
| 2015– | Number of calving events | 322 | 162 | 34 | 9 | 1 | 528 |

| 2016 | Calved area | 134.0 | 515.4 | 1116.8 | 3058.3 | 893.9 | 5718.4 |
| | Area ratio | 2.3% | 9.0% | 19.5% | 53.5% | 15.6% | - |
| 2016–2017 | Number of calving events | 210 | 167 | 50 | 6 | 1 | 434 |
| | Calved area | 91.2 | 563.0 | 1478.2 | 1077.9 | 6141.0 | 9351.4 |
| | Area ratio | 1.0% | 6.0% | 15.8% | 11.5% | 65.7% | - |
| 2017–2018 | Number of calving events | 361 | 145 | 21 | 2 | 0 | 529 |
| | Calved area | 135.4 | 460.1 | 516.8 | 409.5 | - | 1521.7 |
| | Area ratio | 8.9% | 30.2% | 34.0% | 26.9% | - | - |
| 2018–2019 | Number of calving events | 316 | 198 | 22 | 5 | 0 | 541 |
| | Calved area | 123.1 | 610.1 | 478.3 | 1717.9 | - | 2929.5 |
| | Area ratio | 4.2% | 20.8% | 16.3% | 58.6% | - | - |

Based on the area ratio between the $<1$ km$^2$ and 1-10 km$^2$ calving and the 1-10 km$^2$ calving rate of 65.6 Gt/yr, we estimated that the $<1$ km$^2$ calving rate of 18.4 Gt/yr. The 1026 Gt/yr "steady-state" calving flux reported by Liu et al. (2015) subtract the $<1$ km$^2$ calving rate of 770.3 Gt/yr and the $<1$ km$^2$ calving rate of 18.4 Gt/yr, the result of 237.3 Gt/yr is the ice shelf growth due to area expansion. Thus, Antarctica is growing in extent.

We also added these discussions in the discussion section: "The trade-off between workload and uncertainty reduction is another consideration in choosing the minimum spatial scale of calving observation. With the calving scale decreasing from 100 km$^2$, the number of annual calving events increases exponentially, which means that the monitoring workload also increases exponentially (Qi et al., 2020). Although direct calving observation has the minimum valid extraction area of 0.05 km$^2$ based on 75-m SAR resolution images (Qi et al., 2020), it is uneconomical to observe calving events of less than 1 km$^2$ using exponentially increasing manual workload to reduce slightly the uncertainty of the total calving-rate estimation. This is why in the present work the calving area and mass of calving events of less than 1 km$^2$ of the Antarctic ice shelves were estimated based on observation-area ratio and direct observation of 1–10 km$^2$ calving events.

The total circum-Antarctic iceberg calving rate of 955.4±51.4 Gt/yr between 2005 and 2020 observed and estimated in the present study is less than the steady-state iceberg calving fluxes of 1,265 Gt/yr estimated by Rignot et al. (2013) and 1,321 Gt/yr estimated by Depoorter et al. (2013), respectively. The steady-state calving flux is the calving flux necessary to maintain an assumed steady-state calving front for a given set of ice thicknesses and velocities along with the ice-front gate (Rignot et al., 2013; Depoorter et al., 2013). Such "flux-gate" calving calculations for the marine-terminating glaciers are suitable. Our estimated calving rate of the marine-terminating glaciers, 166.7±15.2 Gt/yr, is very close to that reported by Rignot et al. (2013), i.e., 176 Gt/yr. However, such "flux-gate" calving calculations for ice shelves are inevitably biased as they underestimate iceberg calving for retreating ice shelves or overestimate it for advancing ice shelves. Our observed average calving rate of 770.3±29.5 Gt/yr from calving events larger than 1 km2 between 2005 and 2020 is slightly greater than the average rate of 755 Gt/yr between 2005 and 2011 (Liu et al., 2015), which is contributed by two distinct high calving rates of 1,398.8 Gt/yr in 2015/16 and 1,832.6 Gt/yr in 2016/17, respectively. The average calving rate of 788.7±36.2 Gt/yr of all of the Antarctic ice shelves between 2005 and 2020 is the sum of 770.3±29.5 Gt/yr and the estimated average calving rate of 18.4±6.7 Gt/yr from calving events less than 1 km2, which is less than the steady-state calving fluxes of 1,089 Gt/yr estimated by Rignot et al. (2013) and 1,026 Gt/yr estimated by Liu et al.(2015), respectively. Thus, the Antarctic ice shelves are growing in extent."

**Comment 2:**

(Page 2-4) What we know about the relationship between iceberg size and relative abundance is that they generally follow a Pareto distribution. This is also described as the Gutenberg-Richter relationship. Here, for example, is one figure from the Åström et al. 2014 paper that's cited in the present manuscript…\

In addition to adding some essential caveats to the 771.1 Gt/yr value presented in the abstract, I suggest including a few sentences in the Discussion section about what isn't captured in this dataset. Such a discussion might even exploit the Gutenberg-Richter relationship. For example, below I've assumed a Pareto distribution for the 1786 calving events in this dataset, and by fitting a line in log-log space I'm able to extrapolate this relationship down to a hypothetical iceberg size of 1 m$^2$. This gives us some insight about how many 1 m$^2$ icebergs are likely to calve each year, and how many 10 m$^2$ icebergs, and how many 100 m$^2$, and so on. By knowing the number of icebergs that likely exist, but are too small to be included in this dataset, we can start to build some intuition for how much calving flux might be missing from the dataset.

Response 2:

This is a good point. Thank you for your suggestion. Åström et al. 2014 demonstrated that the probability of calving events obeys a particular pattern no matter if they are small or large events—much like the Gutenberg-Richter law for earthquakes. It was interpreted with a catalog of observations that spans 12 orders of magnitude, with data from Alaska, Svalbard, Greenland, and Antarctica (Fig. 2b from the Åström et al. 2014 paper). The author of this article, Yan Liu, co-authored the Åström et al. 2014 paper and collected, processed or interpreted the calving observations calving volumes range from $10^8$ to $10^{12}$ m$^3$ of Antarctic ice shelves between 2005 and 2011. Effective smaller scale calving observation requires higher resolution remotely sensed imagery.

[Figure]

We also added the related discussion: "The interpretation of calving records spanning 12 orders of magnitude from 1 to $10^{12}$ m$^3$ have demonstrated that the probability of calving events obey a particular pattern whether they are small or large events—much like the Gutenberg-Richter law for earthquakes (Åström et al., 2014). Thus, the fine-scale and continuous observation of calving can be used to investigate how close particular glaciers are to their critical point, and thus how sensitive they may be to near-future changes in climatic and geometric conditions. However, finer-scale direct observation is greatly limited by the accessibility of high-resolution remotely sensed imagery and significant manual overhead. Our observations provide records of calving volumes ranging from $10^8$ to $10^{12}$ m$^3$ of Antarctic ice shelves."

**Comment 3:**

(Page 5) Section 3.3.3 is relatively innocuous, but it's unclear what value there is in this type of subjective binning. The thresholds were not set by any sort of natural clustering in the size-vs frequency distribution, and no evidence is given that there is any meaningful glaciological distinction between behavior of "low-frequency" and "high-frequency" calving events. Accordingly, the parameters in Table 3 seem somewhat arbitrary, and I'm not sure what can be gained by classifying a calving event as either "low-frequency" or "high-frequency". I recommend either removing this section from the paper or expanding a little bit on why the bins are meaningful and how they might be interpreted in a glaciological or oceanographic context.

Response 3: Thanks for your suggestion. We have removed the classification of "low-frequency" and "high-frequency" calving events for it seems confused to other readers.

**Comment 4:**

Given that this is a data paper, I suggest including a genuine discussion about what insights can be obtained directly from the data or what benefits are gained by any new methods presented here. To be clear, a list of characteristics of the data is not very insightful on its own. Simply stating that Antarctica calves 127.6 icebergs per year does little for readers without any discussion of why that number is important. A sentence that lists the years of elevated calving flux is somewhat meaningless without talking about why calving rates are high some years or whether interannual variability is dominated by a few large calving events or by broadband increases in calving. Likewise, if there is some sort of insight to be gained by knowing that the formal estimate of calved area uncertainty is 17.1 $km^2$, then by all means, discuss that here. Does the area uncertainty represent some limitation of the dataset? Discuss that here. Can the measurement uncertainty be used to offer readers any words of caution about how to interpret the data? Discuss that here. Can we use these 14 years of observations to understand calving processes that occur on multi-decadal timescales? Discuss that here. If there's anything else that feels important to understanding this data, then please discuss it here.

Response 5: Thanks for your suggestion. We have rewritten the discussion part and added the above content.

**Comment 6:**

**Missing coastline shapefiles:** A handful of papers have been published recently, each claiming to have mapped the Antarctic coastline at annual resolution, but to my knowledge, no such data has been made publicly available by any group. Given the aggressive open-data policy of ESSD and my own personal interest in obtaining such a dataset, I feel compelled to ask about the whereabouts of the annual coastlines that were developed to generate this calving flux dataset. Will the annual coastlines be made available?

Response 6: Thanks for your suggestion. As an intermediate product of calved area extraction, we mapped the frontal edge of Antarctic ice shelves at annual resolution. It's different from the coastline product, for it only portrays the position of the **ice-shelf frontal edge** from year to year (as the examples shown in the figure below), and it directly reflects the change in the area of each ice shelf under the assumption of unchanged grounding line position. We will make the annual coastlines of the ice-shelf frontal edge available when the paper is accepted.

[Figure]

**Comment 7:**

**File format:** The dataset is currently made available as a .rar compressed file. That's a somewhat uncommon format, at least in the United States, and it required me to download special software to decompress the file. Users may experience less of a barrier if instead the data are zipped up in an ordinary, open-format .zip file.

Response 7: Thanks for your suggestion. We have made a new version compressed in .zip format. You can get the updated dataset from the National Tibetan Plateau Data Center (http://data.tpdc.ac.cn/en/), and entitled "Annual iceberg calving dataset of the Antarctic ice shelves (2005-2020)" with DOI: 10.11888/Glacio.tpdc.271250

**Comment 8:**

**Polygon type:** I experienced a very minor issue that I could not read the shapefile data in Matlab, because the polygons were saved as PolygonM or PolygonZ format rather than simple Polygon format. To get around this, I had to open each shapefile in QGIS and re-save as Polygon format before I could open them in Matlab. I work with a lot of shapefiles in Matlab, but this is the first time I've encountered this particular issue. I'm not sure if the inability to read PolygonM or PolygonZ format is specific to Matlab, but it may help more people use the data if it's saved as a plain Polygon format.

Response 8:

Thanks for your suggestion. We have made a new version that can be easily reading in Matlab. You can get the updated dataset from the link in Response 7.

```
>> map=shaperead('C:\Users\lenovo\Desktop\Annual_Calving_2005to2020\Annual_Calving_2005to2020_AIS_V2.shp')

map =

1975x1 struct array with fields:
    Geometry
    BoundingBox
    X
    Y
    ID
    YEAR
    Perimtr_KM
    AREA_KM
    SCALE
    THICKNES_M
    VOLUME_KM
    MASS_GT
    RECURRANCE
    UA_KM
    UH_M
    UC_KM
    REGION
    ICESHELF
```

**Comment 9:**

**Named icebergs and known collapse events:** A few well known calving events occurred during the study period, but they can't be directly queried in the shapefile data. It would be helpful if the attributes of the shapefiles contained iceberg names and approximate dates of major events such cases as the Mertz Glacier tongue calving event of 2010, or iceberg A68 at Larsen C in July 2017, or the successive collapse events at Wilkins Ice Shelf.

Response 9:

Thanks for your suggestion. We have made a new version with the name of the specific ice shelves where each calving occurred. The annual calving events are different from the calved icebergs, for it may consist of many single calving happened in the same year (see as the figure below), therefore we could not match the name of the iceberg with every annual calving events.

[Figure]

**Comment 10:**

**Line 28** is problematic as it currently reads, "In total, 1786 annual calving events occurred on the Antarctic ice shelves from August 2005 to August 2019." The wording of this sentence could easily be misinterpreted, because it says without

qualification that the *total* number of calving events during the study period was just 1786. No doubt this is an underestimate, likely by an order of magnitude or more. For example, the previous paper by Qi et al. reports that 2032 calving events were detected just within the final four years of the present study's period of investigation. I suggest a very simple fix, which is to say something like "we detect a total of 1786 calving events larger than 1 km2 ..."

Response 10: Thanks for your suggestion. We have made the correction.

**Comment 11:**

**Line 187-192:** Calving area uncertainty is also influenced by velocity uncertainty, and should probably be accounted for here.

Response 11:

Thanks for your question. We have quantitatively described the error introduced by the velocity in the paper before (see Qi et al., 2020)., in which the advantages and errors of the calving event extraction method are systematically demonstrated. The estimation of calving area uncertainty has already considered the influence of velocity uncertainty. In the calving area detection, we had a step to move back the detected calving

area by simulated coast to its location before calving and modified the calving boundary difference due to velocity uncertainty. Then the calving area uncertainty is dependent on its length of the boundary. i.e., the perimeter.

**Comment 12:**

**Table 4** took me a while to understand, mainly because I was thrown off by the unitless use of the word *frequency*. The usage of the word frequency is not incorrect, per se, but it is slightly more difficult to parse than simply discussing the number of events that were counted. I think the title should be something like "Number of calving events detected in MODIS and SAR." Likewise, the "Calving frequency" row should be renamed "Number of calving events". It's not immediately clear what is meant by the row labeled "Calving area". Does it mean "Total calved area"? The Scale column contains only text descriptors of Area categories that were binned

based on somewhat arbitrary thresholds, and to interpret them in this table the reader is tasked with going back to Table 3 to figure out what these categories mean. I recommend removing the subjective labels like "Medium-scale" and "Extra-large-scale" throughout the paper and replacing with the area range.

Response 12: Thanks for your suggestion. We have made the correction.

Table 4: Frequency and area distribution of different scale calving events derived from MODIS and SAR for 2016/17

|  | Scale | MODIS | SAR | Δ(MODIS-SAR) | Δ(MODIS-SAR)/SAR$_{Total}$ |
|---|---|---|---|---|---|
| Number of calving events | Small-scale (< 10 km$^2$) | 163 | 167 | -4 | -1.8% |
|  | Medium-scale (10-100 km$^2$) | 50 | 50 | 1 | 0.4% |
|  | Large-scale (100-1,000 km$^2$) | 6 | 6 | 0 | - |
|  | Extra-large-scale (>1,000 km$^2$) | 1 | 1 | 0 | - |
|  | Total | 220 | 224 | -4 | -1.8% |
| Total calved area (km$^2$) | Small-scale (< 10 km$^2$) | 511.0 | 563.0 | -52.0 | -0.6% |
|  | Medium-scale (10-100 km$^2$) | 1,441.0 | 1,478.2 | -37.2 | -0.4% |
|  | Large-scale (100-1,000 km$^2$) | 1,057.9 | 1,077.9 | -20.0 | -0.2% |
|  | Extra-large-scale (>1,000 km$^2$) | 6,054.7 | 6,141.0 | -86.3 | -0.9% |
|  | Total | 9,064.6 | 9,260.2 | -195.5 | -2.1% |
| Standard deviation of total calved area (km$^2$) | Small-scale (< 10 km$^2$) | 2.3 | 2.2 | 0.1 | 0.0 |
|  | Medium-scale (10-100 km$^2$) | 21.3 | 17.9 | 3.4 | 0.2 |
|  | Large-scale (100-1,000 km$^2$) | 93.4 | 91.9 | 1.5 | 0.0 |
|  | Extra-large-scale (>1,000 km$^2$) | 0.0 | 0.0 | 0.0 | - |
|  | Total | 397.2 | 402.8 | -5.6 | -1.4% |

**Comment 13:**

**Section 5.1** contains several uses of the phrase *calving frequency* to mean the total number of calving events that were detected. The distinction may just be a minor wording preference of mine, but it feels significant as the *calving frequency* implies an intrinsic property of the system, whereas the *number of detected calving events* is unambiguous and directly describes what has been measured. For clarity, I recommend replacing *calving frequency* throughout the paper. This would also make room for another phrase used in this paper, *calving recurrence interval*, which might be easily confused with *calving frequency* if both phrases are used in the same manuscript.

Response 13: Thanks for your suggestion. We have made the correction.

**Comment 14:**

**Figure 5** is interesting, but for I would change the phrase "calving frequency" to "number of calving events", change the phrase "calving area" to "total calved area", remove the entire right-hand column, as the colored bins of area shown in the left-hand plots are a natural proxy for frequency.

Response 14: Thanks for your suggestion. We have made the correction.

[Figure]

Figure 5: Temporal distribution of annual calving events at different scales of Antarctic ice shelves from August 2005 to August 2020. Panels (a), (b), and (c) present the annual number of calving events, calved area, and calved mass at four scales, respectively. Horizontal dashed lines in Panel (c) denote the 1026 Gt/yr "steady-state" calving flux of ice shelves reported by Liu et al. (2015)

**Comment 15:**

**Section 5.3** dives into the recurrence interval data, but the concept of the recurrence interval hasn't been adequately described. A fair attempt was made to give sort of a dictionary definition of the term on Line 215, but then by Line 216 the meaning gets lost in vague terms about calving that occurs in the same neighborhood, which depends on some threshold of distinguishing between in-neighborhood and out-of-neighborhood events, and by now I've lost any sense of what the recurrence interval can tell us.

After Section 3.3.3 in which the quantification method is described, the topic of recurrence intervals is comes up again in Section 5.3 without any conceptual bridge to help readers understand physically what is meant by this sentence that begins on Line 307:

*Calving events with a recurrence interval of 3 had the highest frequency, ...*

Already I'm confused, as I have no physical intuition for what the recurrence interval tells us about glaciers, or if this is just a characteristic of the detection limits of the method. Without setting the stage for understanding what the recurrence interval actually tells us, readers are likely to left wondering how once-every-three-year calving events can have a higher frequency than annual calving events. After hitting this

confluence of conceptual roadblocks so early in the sentence, it's difficult then to understand the remainder of the sentence, which continues,

*...accounting for 18.8% of the total and occurring 335 times in 14 years, followed by those with recurrence intervals of 5, 2 and 4, accounting for 18.5%, 15.2% and 14.3%, respectively.*

Again, what do any of these numbers mean? My interpretation is that the methods presented in this paper may not fully capture the types of calving events that occur more often than every three years, because the high-frequency events are more likely to be smaller and go undetected. If that's the case, that's absolutely fine, but explore the concept further, so the folks who use this data will understand what it tells us and what the limitations are. If, on the other hand, there's something glaciologically meaningful here, then please help readers understand it.

…

**Figure 6** is difficult to interpret. I know it's supposed to be communicating something, but I keep gazing into the figure, hoping it will reveal its secrets to me. After some inspection I see that area and mass well correlated, but that is to be expected. And it appears that high numbers of the small icebergs combine to represent a fair portion of the total detected mass, but still

I'm not sure what the take-home message of this figure is.

I think the units of the horizontal axis are supposed to be years, but there are no data in the 1-year bin, so I must wonder,

Are there truly *no* places in Antarctica where calving occurs every year, or is this just a limitation of the detection method? Or am I misunderstanding the meaning of recurrence interval?

I also suspect there's some intention behind the shading of the left and right side of the figure, but I can't figure out what it's trying to communicate.

The task of interpreting this figure is not made any easier by the caption, which contains only a verbless sentence fragment. From the main body of this manuscript, it is clear that the authors are excellent writers, so by all means, use your talents to fill this space with vivid descriptions of what's happening in the figure! Help me understand what I should be noticing in the graphic, and help me understand why it's important.

Response 15:
Thank you for your comments. We modified the description of calving recurrence interval.
Section 3.4.3: "Most calving events are thought to be part of the natural cycle of advance and retreat of ice shelves. Calving recurrence means that a calving event with the same spatial scale reoccurs at the same calving front. The recurrence interval of a calving event is defined as the year interval between the two recurrence calving events."
Section 5.4: "The recurrence interval of calving provides additional qualitative information about the style of calving (Liu et al., 2015) and determines the suitable observation period for identifying ice shelf nonsteady-state behavior. For example, the rift-opening calving of the Amery Ice Shelf has reoccurred in 2019 since the last calving in 1963/64 (Li et al., 2020), detach along the boundary of isolated pre-existing rifts for decades. The observational records spanning many decades would be needed to determine its nonsteady-state behavior. In contrast, more frequent disintegration calving events are mainly caused by the hard to observe rapid basal crevasse propagation (Liu et al., 2015). The calving front retreat associated with these frequent calving events can be robustly identified over a short observation period due to the shorter recurrence intervals. In other words, the calving events with shorter recurrence intervals are more sensitive to current climate change.

Figure 6 (a) shows the calving recurrence interval is little related to calving scales of caving. The two extra-large-scale (> 1,000 km$^2$) calving events reoccurred on the Thwaite Glacier during our observed period indicating its distinct retreat, while the other four extra-large-scale events from the Larsen C, Wilkins, Totten, and Amery Ice shelves did not reoccur. Figure 6 (b) shows that 76% of the total number of calving events reoccurred during the observed period (i.e., their recurrence intervals of calving are less than 8 yr), which suggests that the annual calving number is likely to be an indicator of the response of calving to climate change. Nearly half of the cumulative calved area from the events with the recurrence intervals greater than 8 yr (i.e., the events only occurred once during the observed period) suggests that the annual calved area is not suitable for identifying the nonsteady-state behaviors of some ice shelves."

We modified Figure 6, see below:

[Figure]

**Figure 6: Distribution of calving events with different recurrence intervals. Panel (a) shows the cumulative number of calving events at different scales. Panel (b) shows the cumulative percentages of the cumulative number of calving events, the cumulative calved area, and the cumulative calved mass.**

**Comment 18:**

**Line 344** says "the Totten Ice Shelf was collapsing every year." The word "collapsing" may be a bit too strong, so consider changing to something like "We detect calving events at Totten Ice Shelf every year."

Response 18: Corrected.

**Comment 19:**

**Lines 350 to 363** rehash numbers from earlier in the manuscript without reframing or adding any new perspectives. I think this paragraph can be deleted without any detriment to the paper.

Response 19: Thanks for your suggestion. We have made the correction and rewritten this section.

**Comment 20:**

ENSO analysis seems out of place: The attribution of calving events to ENSO anomalies and surface melting looks very interesting, but the analysis feels out of place in this data paper, particularly as the topic is only introduced in the final closing remarks of the paper. If these scientific results are compelling as they appear to be, then they should be described in detail in a separate paper, where they can be discussed at length

while being given a chance for proper peer review.

**Lines 363 to 373** convey a level of enthusiasm that should absolutely be harnessed to develop this analysis further. The early results look very interesting indeed, but I don't think the Discussion section of a data paper is the appropriate place to present new scientific findings about correlations between ENSO and iceberg calving. I recommend removing these two paragraphs and Figure 9 from the paper.

Response 20:

Thanks for your suggestion but we prefer to keep this part.

Our original intention of producing a long time series of fine observation datasets was to analyze the characteristics of calving in the background of climate change. The most important advantage of our annual observed calving rather than the steady-state calving flux is that it can reflect the calving response to climate change. The correlations between ENSO and iceberg calving demonstrate this point. It also demonstrates that the dataset can be used to investigate the external atmospheric and oceanic impacts on iceberg calving. This is a data description paper, we will concentrate more on the description of the data itself. In the discussion section we show a preliminary application of this data, that is, the attribution of calving events to ENSO anomalies and surface melting. We are also conducting in-depth research and hope that will be given a chance for peer review in the upcoming manuscript.

---

## Referee Report (RR1)

Comments for ESSD-2020-340 (revised manuscript)

The authors have addressed all the issues in the revised manuscript.

Minor comments:

1. In response 2, my understanding is that except for the actual coastline in 2005, 2010, and 2015 which were extracted from images (checked and manually corrected), the actual coastline in given year was modified from the simulated coastline by comparing with the images in that year. Considering that only the calved area was modified, so the cumulative error of the non-calved area will be larger after several years and a new benchmark (years of 2005, 2010 and 2015) is needed. I would like to know if there is any basis for dividing 15 consecutive years into three intervals, such as how much error is there before a new benchmark is needed.

2. Line 319, please add the unit "Gt" into calved mass.

---

## Author Response (AR2)

College of Global Change and Earth System Science

Beijing Normal University

No. 19, Xinjiekouwai Street, HaiDian District

Beijing 100875, China

29/5/2021

Dear Editors,

**Revision of ESSD manuscript 2020-340:**

**A 15-yr Circum-Antarctic Iceberg Calving Dataset Derived from Continuous Satellite Observations**

It was a pleasure for us to read the encouraging comments and important suggestions provided by the reviewers. They were of great assistance in improving the quality of the manuscript. All of the comments and suggestions were considered during the revision process. Moreover, we have provided a new version of the manuscript along with a point-by-point response to all of the reviewers' comments.

The following pages provide the point-by-point responses to the suggestions made by the editor and reviewers and a detailed description of the changes made.

Yours sincerely,

The authors

**Minor comments:**

1. **In response 2, my understanding is that except for the actual coastline in 2005, 2010, and 2015 which were extracted from images (checked and manually corrected), the actual coastline in given year was modified from the simulated coastline by comparing with the images in that year. Considering that only the calved area was modified, so the cumulative error of the non-calved area will be larger after several years and a new benchmark (years of 2005, 2010 and 2015) is needed. I would like to know if there is any basis for dividing 15 consecutive years into three intervals, such as how much error is there before a new benchmark is needed.**

   Response 1:

   In order to reduce the error caused by the ice velocity during the iterative calculation of the coastline, we divided 14 consecutive years into three intervals. We used coastlines of 2005, 2010, and 2015 (checked and manually corrected) as the benchmark to control the error of coastline simulation within five years.

   We have evaluated the positional accuracy of the simulated coastline in one-year interval (see Qi et al., 2020). The samples from the ice shelves' frontal edges only advanced, but those without calving events were chosen to assess the accuracy of the simulated coastline (as shown in the Figure 1). The specific method used is as follows. First, calculate the distance from the moved feature points on the simulated coastlines to the automatically extracted, and manually corrected, pixel boundaries of the ice shelves and the sea in the images. Then, determine the directions. If the feature points fall within the sea, the positional error is the distance value above. If the feature points fall on the ice shelf, the positional error is the negative value of the distance.

   We measured the positional errors of all of the feature points (752 in total) in 30 samples. The errors generally exhibit a normal distribution with a standard deviation of 74.6 m and a mean value of 27.9 m. To some extent, there is a systematic error. This is because we regarded the distance from the feature points to the edge of the raster as the positional error, when in fact, the edge pixels are usually mixed pixels of ice and water. To further improve the accuracy, we choose the three benchmark coastlines (years of 2005, 2010 and 2015) as input, this is a choice that balances both the positional accuracy and the workload.

[Figure]

Figure 1. Samples used to validate the positional accuracy of the simulated coastline. The purple lines are the benchmark coastlines of the ice shelf's frontal edge in 2015. The blue points and yellow points are feature points before and after moving based on the ice velocity, respectively. The red lines are the simulated coastlines in 2016 generated by sequentially connecting the yellow points.

(Qi, M., Liu, Y., Lin, Y., Hui, F., Li, T., and Cheng, X.: Efficient Location and Extraction of the Iceberg Calved Areas of the Antarctic Ice Shelves, Remote Sens., 12, 10.3390/rs12162658, 2020.)

**2. Line 319, please add the unit "Gt" into calved mass.**

Response 2: Corrected.